# Hypothetical Minds: Scaffolding Theory of Mind for Multi-Agent Tasks with Large Language Models

**Logan Cross**[1*]   **Violet Xiang**[2]   **Agam Bhatia**[1]   **Daniel L.K. Yamins**[1,2]   **Nick Haber**[3]

Stanford University
[1]Department of Computer Science
[2]Department of Psychology
[3]Graduate School of Education

## Abstract

Multi-agent reinforcement learning methods struggle with the nonstationarity of multi-agent systems and fail to learn online when tested with novel agents. Here, we leverage large language models (LLMs) to create an autonomous agent that can handle these challenges. Our agent, Hypothetical Minds, consists of a cognitively-inspired architecture, featuring modular components for perception, memory, and hierarchical planning over two levels of abstraction. We introduce the Theory of Mind module that scaffolds the high-level planning process by generating hypotheses about other agents' strategies in natural language. It then evaluates and iteratively refines these hypotheses by reinforcing hypotheses that make correct predictions about the other agents' behavior. Hypothetical Minds significantly improves performance over previous LLM-agent and RL baselines on a range of competitive, mixed motive, and collaborative domains in the Melting Pot benchmark, including both dyadic and population-based environments. Additionally, comparisons against LLM-agent baselines and ablations reveal the importance of hypothesis evaluation and refinement for succeeding on complex scenarios.

## 1 Introduction

A fundamental challenge in artificial intelligence is creating agents that can rapidly adapt to others in multi-agent environments. Whether in competitive settings like games, collaborative tasks like team projects, or mixed-motive scenarios like negotiation, success requires building predictive models of another agent's behavior by inferring their hidden strategies and latent capabilities. One approach involves training reinforcement learning models that can learn adaptive policies. However, multi-agent reinforcement learning (MARL) methods suffer from various drawbacks, including high sample complexity, poor generalization to agents not seen in training, and limited reasoning capabilities (Wong et al., 2023).

Another approach deploys foundation models as the backbone to agents, with specialized modules that mediate the decomposition of long horizon planning (Wang et al., 2024; 2023a; Brohan et al., 2023a; Park et al., 2023). LLMs are not only powerful reasoners and in-context learners, but they are also particularly suited for social tasks given the utility of language for scaffolding Theory of Mind (ToM) (Astington & Baird, 2005; de Villiers, 2007; 2021; Kosinski, 2023; Gandhi et al., 2024). Thus, we introduce Hypothetical Minds (HM), a LLM agent that produces adaptive policies in competitive, cooperative, and mixed-motive multi-agent scenarios. At its core, HM addresses the challenge of interacting with agents whose behavior is driven by latent features - first inferring these features through a natural language approximation of Bayesian inference, then conditioning its policy on these inferences.

---

*Corresponding author: `locross@stanford.edu`

Code: https://github.com/locross93/Hypothetical-Minds/

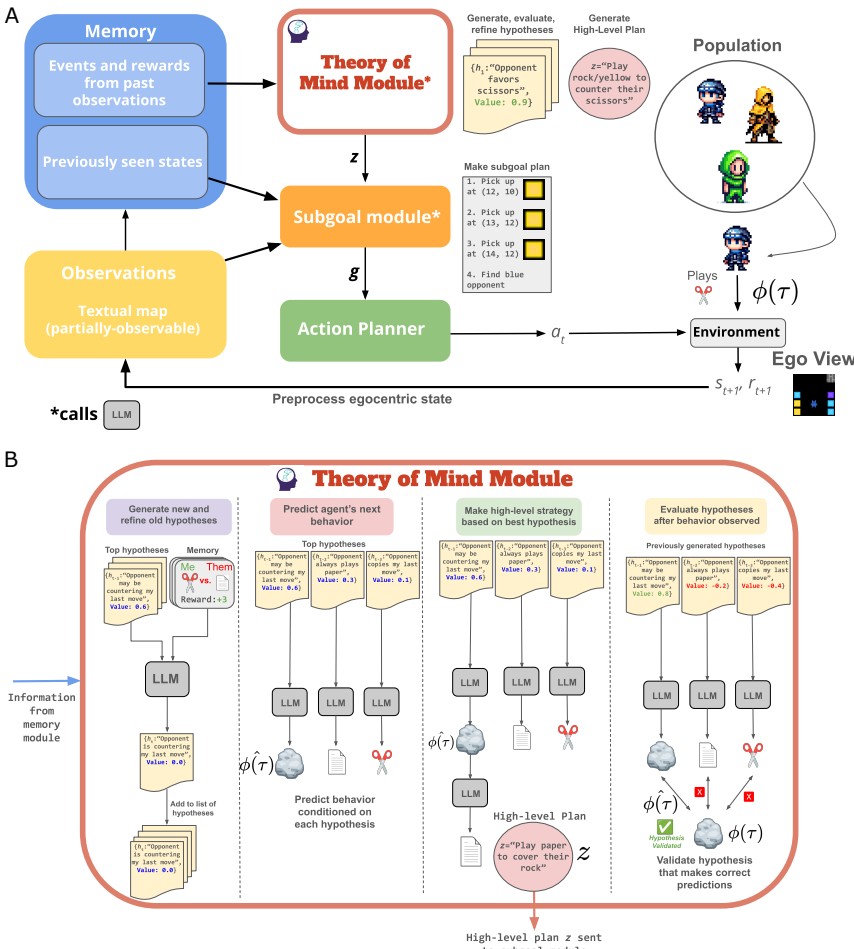

Figure 1: A. **Hypothetical Minds** architecture and model workflow. B. **ToM module** generates hypotheses about agent strategies. Previously generated hypotheses and values are shown for refinement. Top k hypotheses predict agent's next behavior $\phi(\hat{\tau})$, considering counterfactual scenarios. Highest-valued hypothesis informs high-level planning. Later, hypotheses are evaluated against observed behavior $\phi(\tau)$, updating values with intrinsic reward. Hypotheses are validated at a threshold.

HM builds on the generative agents architecture, integrating modules for perception, memory, and hierarchical planning with novel ToM machinery (Park et al., 2023). The Theory of Mind module generates and evaluates hypotheses about other agents' strategies, scoring them based on their ability to predict future actions. Thus, our agent finds useful explanations of other agents' behaviors in-context, affording it the ability to adapt to the inferred strategies and achieve high rewards.

We evaluate our model on the Melting Pot MARL benchmark that encompasses diverse challenges and social dynamics (Agapiou et al., 2022). Collaborative Cooking Asymmetric requires effective coordination and division of labor among agents. Running With Scissors necessitates strategic reasoning about opponents' policies and the ability to exploit predictable patterns in a competitive setting, with the eight-player version offering a unique challenge to the model's scalability. Moreover, the mixed-motive environment Prisoner's Dilemma involves a tension between individual and collective interests. Diverse evaluation scenarios stress test playing with a wide array of agents with fixed or adaptive policies, necessitating contextual adaption. Hypothetical Minds surpasses LLM agent baselines on a majority of evaluation scenarios in every environment, and performs better than MARL baselines on 3/4 environments despite the fact that those methods are trained on upwards of a billion steps. To our knowledge, Hypothetical Minds is the first LLM agent for diverse multi-

agent environments, with strong performance across collaborative, competitive, and mixed-motive domains, unlike previous works limited to collaborative settings. Our contributions are as follows:

- We propose the Hypothetical Minds model (HM), an embodied LLM agent for multi-agent environments that integrates modular components for perception, memory, and hierarchical planning conditioned on ToM inferences.

- HM incorporates a novel Theory of Mind (ToM) module, which generates, evaluates, and refines hypotheses about other agents' strategies or goals in natural language. Through ablations and comparisons against LLM-agent baselines, we identify the critical role of hypothesis evaluation and refinement within the ToM Module.

- We demonstrate the effectiveness of HM Minds across multiple multi-agent environments, including competitive, collaborative, and mixed-motive domains, and 30 distinct evaluation scenarios. Our agent significantly outperforms LLM-agent and RL baselines in every environment and in a large majority of evaluation scenarios, showcasing its generalizability.

## 2 RELATED WORK

### 2.1 LLM-BASED AGENTS

A burgeoning area of research involves building autonomous agents rooted in large language models Wang et al. (2024); Sumers et al. (2023). This involves deploying LLMs as central controllers across many different domains by leveraging their extensive background knowledge from training. Applications span a wide range from equipping LLMs with external tools to interface with databases and APIs (Schick et al., 2024; Shen et al., 2023; Qin et al., 2023) to using them for high-level planning and control in robotics (Huang et al., 2022; Brohan et al., 2023b; Rana et al., 2023; Brohan et al., 2023a). The most relevant branch of this research direction includes works where LLMs are used as planners in embodied virtual environments. Voyager autonomously builds complex skills in Minecraft by storing and retrieving behaviors in a skill library of executable code and uses the skill library to solve progressively harder tasks (Wang et al., 2023a; 2024). In this work, we use an LLM for long horizon high-level planning and predicting the future states of other agents in multi-agent environments. Previous papers have also incorporated LLM-based agents into embodied multi-agent environments (Liu et al., 2023a). Park et al. (2023) introduce a interactive simulation of a rich social environment, where each agent autonomously selects goals and builds relationships with others. We extend the cognitive module framework developed in this work for multi-agent environments of varied dynamics by introducing the novel Theory of Mind module. SAMA uses an LLM to plan out sequences of subgoals for language-based goal-conditioned RL policies in environments requiring multi-agent coordination Li et al. (2023). Another study builds cooperative embodied agents, by using an LLM for planning and communication between agents Zhang et al. (2023b). ProAgent develops a method for improving zero-shot coordination in Overcooked by using an LLM to infer the intentions of teammates based on the present state Zhang et al. (2023a). As these works have focused solely on collaborative domains, here we present a generalizable and scalable method that addresses the challenge of inferring other agents intentions' across a wide spectrum of social dynamics.

### 2.2 REASONING AND HYPOTHESIS SEARCH WITH LLMS

LLMs have shown impressive reasoning abilities, augmented by Chain-of-Thought methods that scaffold the thought process (Wei et al., 2022; Zhang et al., 2023c). Our hypothesis generation approach builds on this work by implementing structured reasoning steps for theory of mind inference and decision-making. Tree-of-Thoughts (Yao et al., 2024) extends chain-of-thought by maintaining and evaluating multiple reasoning paths in parallel, similar to our parallel evaluation of multiple hypotheses about opponent strategies. ToT evaluates different solution paths of a reasoning trajectory with an external verifier or LLM as judge, whereas we score hypotheses about an agent based on the quality of predictions they make about that agent's behavior. The structured prompting in our ToM module draws on broader findings about effective prompt engineering for complex reasoning (Zhang et al., 2023c). For example, our separation between reasoning and action in our prompts draws inspiration from the ReAct framework (Yao et al., 2023).

Recent work has specifically investigated LLMs' capabilities for hypothesis generation and testing. Wang et al. (2023b) explores LLMs' inductive reasoning by generating and evaluating hypotheses on the Abstraction and Reasoning Corpus (ARC), while Qiu et al. (2023) refines LLM-generated

hypotheses using task-specific symbolic interpreters. Similarly, we generate, evaluate, and refine hypotheses based on feedback, though in our case computing values for each hypothesis by predicting another agent's goals. STaR (Zelikman et al., 2022) also learns from feedback by finetuning language models on rationales that produced correct answers.

## 2.3 MULTI-AGENT DECISION-MAKING AND THEORY OF MIND

Decision-making in multi-agent settings has been extensively studied through multi-agent reinforcement learning (MARL), with benchmarks like Melting Pot providing diverse social interaction scenarios to evaluate agents (Agapiou et al., 2022). While algorithms like MADDPG (Lowe et al., 2017) and MAPPO (Yu et al., 2022a) have advanced MARL through centralized training approaches, specialized models incorporating Theory of Mind principles (Rabinowitz et al., 2018; Wang et al., 2021; Yu et al., 2022b; Huh & Mohapatra, 2024; Vezhnevets et al., 2020; Sclar et al., 2022) have shown promise in modeling other agents' intentions. However, even sophisticated approaches like OPRE (Vezhnevets et al., 2020) struggle to generalize in complex environments like Melting Pot's Running With Scissors (Agapiou et al., 2022), highlighting MARL's limitations in compute requirements and training stability. In cognitive science, the Rational Speech Act framework (Goodman & Frank, 2016) and Bayesian inverse planning (Baker et al., 2009) model how humans infer others' goals and beliefs through Bayesian inference, providing inspiration for our ToM module.

## 3 METHOD

### 3.1 PARTIALLY-OBSERVABLE MARKOV GAMES

Our method is directly applicable to any multi-agent environment where states are partially observable and agent(s)' policies are hidden. We formally define this as a Markov game for $N$ players in a partially observable setting. Let the finite set $S$ represent the possible states of the game. Each player $i$ receives observations given an observation function $\chi^i : S \to \mathcal{O}$, representing their limited point of view. Additionally, each player $i$ can take actions from their action space $A^i$, and when all players choose actions $(a^1, \ldots, a^N) \in A^1 \times \cdots \times A^N := A$, the state transitions according to a probability distribution $T : S \times A \to \mathcal{D}(S)$. The reward function for each player $i$ is represented as $r^i : S \times A \to \mathbb{R}$, mapping the current state and joint actions to a real-valued reward.

### 3.2 HYPOTHETICAL MINDS MODEL

HM consists of several cognitive modules that altogether form an embodied LLM agent (Figure 1). The egocentric observations are represented by a textual map/state representation, which is added to a memory system after every step. The memory system also logs rewards and other important state information like the inventories from previous interactions in Running With Scissors. Two cognitive modules depend on an LLM, a Theory of Mind module and a Subgoal module, which output high-level goals and action plans respectively. An action planner takes an action plan (i.e. "move to coordinate (13, 5)") and creates a sequence of actions that achieves that action plan with a pathfinding algorithm. Each cognitive module is explained in more detail in the Appendix. Below we describe the key novel contributions of our method that implement hierarchical planning.

**Theory of Mind Module** In multi-agent environments, other agents' behavior can be influenced by various latent variables, such as their strategies (e.g., "always defect"), goals (e.g., "trying to pick up an object"), competence levels (e.g., "highly skilled" vs. "negligent") and locations in an environment. These latent variables are often not directly observable and must be inferred from partially-observable information from your perspective as the ego agent. We represent these latent variables as a multidimensional space $\Theta = \theta_1, \theta_2, \ldots, \theta_m$, where each dimension $\theta_i$ corresponds to a specific latent variable in an embedding. The ToM module generates hypotheses about these latent variables in natural language. It then maintains a set of hypotheses $\mathcal{H} = h_1, h_2, \ldots, h_n$ in its working memory, where each hypothesis $h_t$ represents a belief about the latent variables $h_i = p(\Theta)$. A hypothesis at time $t$ is generated by asking an LLM to infer another agent's strategy, conditioned on HM's memory $\mathcal{M}$ of important past observations $\mathcal{O}$. Additionally, the LLM is shown the top k valued previously generated hypotheses, such that it can perform **hypothesis refinement** (see Appendix and code for more details and prompts):

$$h_t = \text{LLM}(\mathcal{M}, h_{<t}) \qquad (1)$$

where $\mathcal{M}$ is a memory buffer storing an agent's past actions, observations, and rewards.

Each hypothesis $h_i$ is scored based on how well it predicts the other agent's future behavior, noted here formally as a distribution of trajectories $p(\tau)$. We formalize this scoring mechanism using a likelihood function $p(\tau|h_i)$ representing the probability of an agent exhibiting trajectory $\tau$ given the hypothesis $h_i$. The best hypothesis $h^*$ is selected using the Maximum a Posteriori (MAP) estimate:

$$h^* = \underset{h_i \in \mathcal{H}}{\arg\max}\, p(h_i|\tau) = \underset{h_i \in \mathcal{H}}{\arg\max}\, \frac{p(\tau|h_i)p(h_i)}{p(\tau)} \tag{2}$$

where $p(h_i)$ is the prior probability of hypothesis $h_i$ and $p(\tau)$ is the marginal probability of the observed action $a$ and has no effect on the argmax. The likelihood is approximated by a hypothesis evaluation mechanism described below. The LLM predicts the other agent's future behavior conditioned on each hypothesis separately. Hypotheses leading to correct predictions will have higher values reflecting higher likelihoods. The prior $p(h_i)$ corresponds both to the background knowledge embedded in the weights of an LLM from pretraining and to the refinement mechanism that shows the top valued hypotheses to the LLM when the LLM generates a hypothesis. By continuously updating and selecting the best hypothesis based on observed information, the ToM module can effectively infer the latent variables governing the other agents' behavior and adapt its own strategies accordingly. When the environment has more than one other agent, the ToM module maintains separate hypothesis streams for each other agent, with unique hypotheses and values tracked per agent.

**Hypothesis Evaluation**   Drawing on cognitive modeling approaches (Rescorla, 1972; Daw & Tobler, 2014), multiple hypotheses are scored with a value system $V_{h_i} = \mathbb{E}[r]$ where $r$ reflects intrinsic reward based on the accuracy of the predictions the hypothesis generates. We compute self-supervised intrinsic rewards bootstrapped from the LLM's own predictions. Let $\phi(\tau)$ be a particular behavior, a feature from an observed trajectory, and $\hat{\phi}(\tau)$ be the predicted behavior by the LLM, such as the inventory played by an agent in Melting Pot environments. The intrinsic reward function $r_i$ can then be defined as:

$$r_i = \begin{cases} c & \text{if } \hat{\phi}(\tau) = \phi(\tau) \\ -c & \text{if } \hat{\phi}(\tau) \neq \phi(\tau) \end{cases}$$

where $c$ is a hyperparameter. $V_{h_i}$ is then dynamically updated with a Rescorla Wagner update rule (Rescorla, 1972) , expressed as:

$$\delta = r_i - V_{h_i}$$

$$V_{h_i} \leftarrow V_{h_i} + \alpha \cdot \delta$$

modulated by learning rate $\alpha$ via a prediction error $\delta$. The learning rate dictates how much to weigh recent interactions, a useful property when playing against evaluation scenarios where agents change their strategy within an episode. When the value of a hypothesis meets a threshold $V_{\text{thr}}$, the ToM module marks the hypothesis as validated and uses this hypothesis to condition high-level plans (using the highest-valued one if multiple pass the threshold). This hypothesis will then continue to be used for planning until it no longer makes good predictions and its value subsequently falls below the threshold. If no hypothesis meets the threshold, by default the latest generated hypothesis (with the most updated information) is used for conditioning high-level plans. The highest value hypothesis could also be used in our code. We set the validation threshold Vthr = 0.7 apriori based on analysis of the Rescorla-Wagner dynamics. With reward values of c = 1 and learning rate = 0.3, a hypothesis requires consistent predictive accuracy to reach and maintain this threshold. The top_k hyperparameter mediates cost trade-offs, representing how many of the top_k old hypotheses are continually evaluated in parallel. Each ToM module call uses (3 + top_k) LLM calls if no hypothesis is validated yet and only 2 calls after validation, thus scaling linearly at the step in computational cost when hypotheses are not validated and sublinearly when they are frequently validated. (See Table 4 in Appendix for all hyperparameters and Table 5 and Appendix Section B for information about computational costs). The default value as reported in the results use top_k = 5.

**Conditioning High-Level Plans**   The ToM module then conditions its high-level plans on the inferred latent variables represented by the hypotheses. A high-level plan $z$ is a natural language

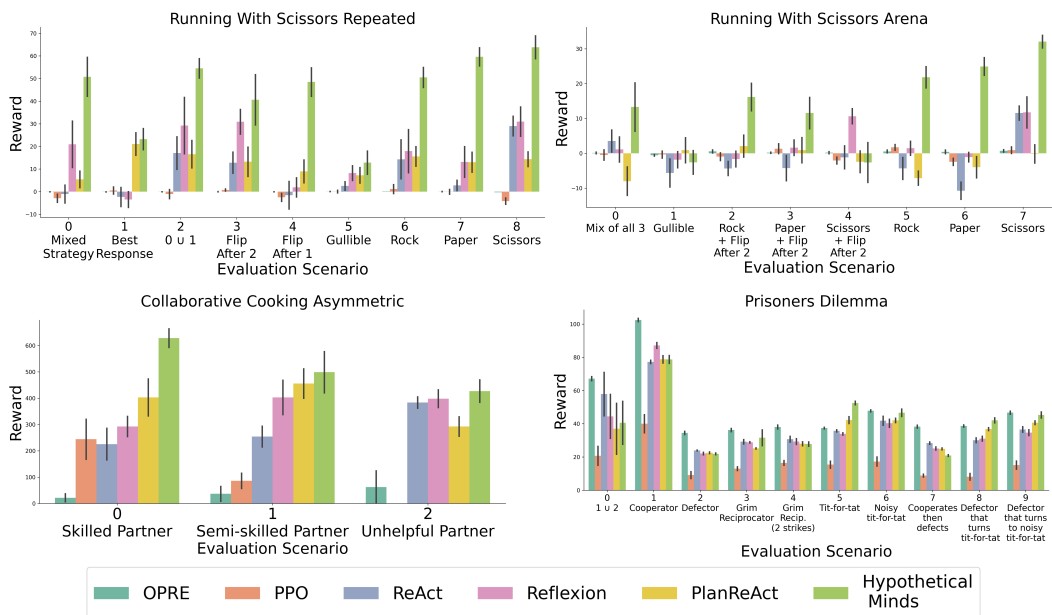

Figure 2: Results for all models. Average reward per episode (with normalized steps for variable length episodes) for each environment and scenario. 5 seeds are generating for each model, with errorbars reflecting the SEM across those 5 episodes.

description of HM's overall strategy, goal, or intention, conditioned on the best hypothesis $h^*$ and memory of past events:

$$z = \text{LLM}(\mathcal{M}, h^*) \tag{3}$$

By conditioning the high-level plans on the hypotheses, HM can adapt its strategy based on its understanding of the other agents' latent states.

**Subgoal Module** Finally, the Subgoal module selects a sequence of subgoals. Let $\mathbf{g} = g_1, g_2, \ldots, g_k$ be a sequence of subgoals, where each subgoal $g_i$ is an action or short sequence of actions that the agent needs to take to achieve the high-level plan $z$:

$$\mathbf{g} = \text{LLM}(\mathcal{O}, \mathcal{M}, z) \tag{4}$$

The sequence is generated by conditioning the LLM on the high-level plan, observations, and memory, and prompt to achieve the high-level plans. The LLM outputs a sequence of specified subgoal function calls, which are then parsed and mapped to the corresponding actions in the environment by a hardcoded Action Planner (see Appendix).

## 4 EXPERIMENTS

Here we investigate the following to analyze the generalizability and scalability of our method:

Q1. How does Hypothetical Minds perform compared to LLM agent and RL baselines in embodied competitive zero-sum environments?

Q2. How does Hypothetical Minds perform compared to LLM agent and RL baselines in a collaborative domain that requires adaptation to a partner's role and competence?

Q3. How does Hypothetical Minds perform compared to LLM agent and RL baselines in a mixed-motive setting?

Q4. Does the Hypothetical Minds agent scale effectively to environments with larger populations of agents?

Q5. How do the different components of the Hypothetical Minds agent and the Theory of Mind module contribute to its overall performance?

We directly test our LLM-based agent on the evaluation scenarios in four Melting Pot environments (Figure 2). The key evaluation method is to test agents against different bots with various policies.

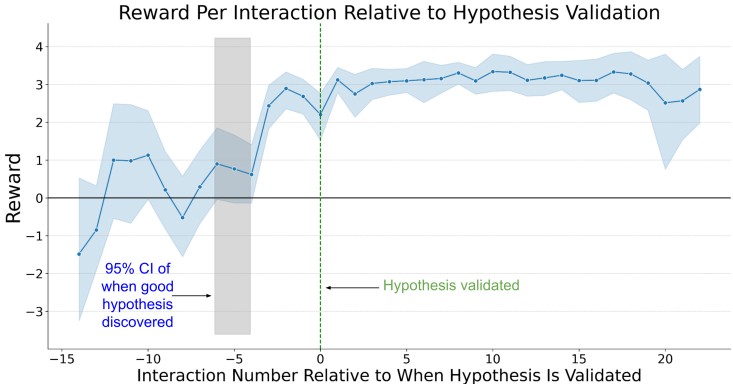

Figure 3: HM's reward per number of interactions before or after a hypothesis meets the validation threshold and is used for high-level strategy selection in RWS. Vertical green line indicates the average reward at the point where a hypothesis is validated, and positive and negative numbers on the x-axis indicate how many interactions before or after this point. Shaded region represents the range where the good hypothesis is typically first generated with a 95% confidence interval.

Across environments, this consists of 30 distinct evaluation scenarios. Crucially, our agent has no knowledge about which strategies they may be playing in the prompts given. Strategies have to be ascertained online within an episode via in-context learning.

**Baselines**

**ReAct** synergizes reasoning and acting in language models, allowing them to generate both reasoning traces and task-specific actions in an interleaved manner (Yao et al., 2023). **Reflexion** includes three main components: an Actor module that generates actions and text, an Evaluator that scores these actions, and a Self-Reflection module that uses the evaluations to provide constructive feedback stored for subsequent use (Shinn et al., 2024). **PlanReAct** To provide a hierarchical baseline to test against our hierarchical model, we include the PlanReAct architecture introduced in (Liu et al., 2023b). This structure allows the agent to plan before interacting with the environment. **PPO** is a model-free RL baseline (Schulman et al., 2017) and we train agents in a population of models with the same parameters. **OPRE** is a hierarchical MARL method where agents learn high-level options as strategic responses to other agents' behaviors (Vezhnevets et al., 2020). The high-level controller selects options based on observations of other agents, and the low-level controller executes actions conditioned on these options. We include the OPRE results from the Melting Pot 2.0 paper (Agapiou et al., 2022) as a baseline to compare against a hierarchical MARL approach that, like our model, aims to adapt dynamically to other agents' strategies.

Prompts and architectures are shared across baselines to provide a fair comparison and natural ablations of our model. The Subgoal module provides a baseline actor shared between LLM baselines, and the only difference between PlanReAct and Hypothetical Minds is that high-level planning is mediated by the Theory of Mind module, including hypothesis generation, evaluation, and refinement.

### 4.1 How does Hypothetical Minds perform in competitive environments?

**Running With Scissors in the Matrix Repeated (RWS)** is a zero-sum competitive environment with two players moving around a map and collecting yellow, purple, or blue resources that correspond to rock, paper, and scissors respectively. Zapping your opponent causes an interaction, with one agent getting positive reward and the other agent getting an opposite negative reward according to the inventories of resources picked up by each player, mirroring the rock, paper, scissors matrix game.

RWS presents nine distinct evaluation scenarios, which range in complexity. These include three straightforward strategies where opponents consistently play rock, paper, or scissors. The remaining scenarios introduce more dynamic and adaptive strategies (see Appendix for details and full list). Therefore in order to succeed on the scenarios, an agent needs to correctly infer the strategy and exploit it. Against the simple policies, the agent should play the same type of inventory every round rather than playing randomly or anticipating a change in its opponent's policy. In contrast, success

against the adaptive strategies demands not only the anticipation of the opponent's next move based on personal previous plays but also selecting the most advantageous counter.

Figure 2 and Appendix Table 1 demonstrate how Hypothetical Minds model consistently achieves large magnitude rewards and performs reliably better than the baselines on every single scenario. Hypothetical Minds performs the best on the static strategies, scenarios 0, 6, 7, 8 representing fixed policies that play for the same inventory on every interaction (6: rock, 7: paper, 8: scissors, 0: random sample from 6-8). Therefore it is able to exploit the static strategy reliably once it correctly infers it. The agent is also able to consistently return positive rewards against the difficult scenario 1, the adaptive bot that plays the best response to your last round. In contrast, baselines are failing to achieve consistent positive rewards. OPRE, a method designed specifically for a earlier version of this task, gets rewards near zero for every scenario. PlanReAct is the only other model to achieve positive rewards on the difficult best response bot, illustrating the usefulness of a hierarchical structure for this evaluation in particular. Reflexion performs second best overall, underscoring the value of evaluative reflection for this environment.

Figure 3 showcases HM's dynamics, depicting the agent's reward per interaction before and after hypothesis validation. Upon validation, the agent consistently achieves high positive returns, while rewards are near zero or negative during the information-gathering phase. The upward trajectory in reward after generating a good hypothesis and the significant increase just before the validation threshold demonstrate the agent's ability to exploit accurate hypotheses effectively.

## 4.2 HOW DOES HYPOTHETICAL MINDS PERFORM IN COLLABORATIVE ENVIRONMENTS?

In the **Collaborative Cooking: Asymmetric** environment, two players on distinct sides of a divided kitchen must collaborate to efficiently cook tomato soup. The layout provides distinct advantages to each side—one side is closer to the goal delivery but farther from the tomato dispenser, and vice versa for the other side. To maximize rewards, the two players should specialize based on their proximity to resources: one handles tomato dispensing and delivery, and the other manages dish retrieval and soup delivery. Evaluation scenarios challenge the focal agent to demonstrate dual forms of adaptation: adjusting to the specialized role dictated by their side of the kitchen and to the varying competence of their partner, from skilled and specialized to entirely unresponsive.

Again, Hypothetical Minds achieves higher rewards than the baselines on every scenario (Figure 2). Interestingly, HM performs significantly better than the baselines on the scenarios where there is a functional partner (Appendix Table 1). This suggests that if there is value in a partner, HM can take advantage of this and adapt its behavior accordingly, highlighting the model's usefulness for complex, dynamic environments where cooperative interaction is crucial. The LLM baselines perform relatively well with the negligent partner, where success hinges on repeatedly executing an intuitive sequence of actions — filling and delivering pots — without the need to take into account the actions of another mind. For instance, baseline LLM agents struggled when encountering a pot already at capacity while attempting to add tomatoes. PPO showed the opposite relative weakness, performing moderately with the skilled partner but has a dramatic lack of generalization to the unhelpful partner scenario, as it is used to role specialization from self play. OPRE performs poorly on every scenario.

## 4.3 HOW DOES HYPOTHETICAL MINDS PERFORM IN MIXED-MOTIVE ENVIRONMENTS?

In the Prisoner's Dilemma in the Matrix Repeated (PD) environment, agents navigate a similar grid world to RWS and collect resources corresponding to cooperation or defection in the iterated prisoner's dilemma game. The payoff matrix incentivizes mutual defection in a single interaction, but the highest total welfare is achieved through mutual cooperation across an episode.

Hypothetical Minds achieves the highest reward among LLM agents and in 5/10 scenarios (Figure 2, Appendix Table 1). This highlights its ability to perceive the background agent's strategy and adapt accordingly. HM outperforms the LLM baselines relatively more with dynamic partners. With tit-for-tat for example (scenarios 5 and 6), Hypothetical Minds achieves the highest score among all models, by engaging in more consistent cooperation while demonstrating some forgiveness to avoid cycles of alternating defection, a pattern that plagues LLM baselines. In scenarios 8 and 9, Hypothetical Minds showcases a capacity for "corrective punishment." By defecting against these partners that initially defect, it persuades them to switch to conditional cooperation. The agent then shifts to a forgiving cooperation strategy to maintain a mutually beneficial equilibrium.

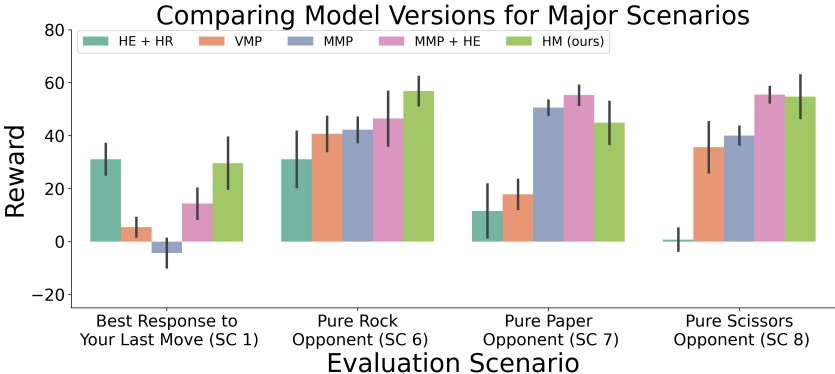

Figure 4: Comparing different versions of HM. Errorbars reflect SEM across 5 episodes.

However, OPRE achieves the highest rewards overall on Prisoner's Dilemma. Fixed heuristic policies such as always defecting, playing tit-for-tat, etc. can achieve high rewards without needing to correctly infer the policy it is playing against. One explanation is that advanced RL methods like OPRE collect resources more efficiently than Hypothetical Minds in Prisoner's Dilemma, due to the limited spatial reasoning abilities of ungrounded LLMs like GPT4. One limitation of HM is that the agent frequently travels across the environment to pick up resources that were closer to its original location. On the other hand, HM demonstrates good reasoning about the strategies it is playing against, and good hypotheses are frequently found and validated. This illustrates the tradeoffs between these two methods. LLM agents provide better abstract reasoning about things like other agent's goals right out of the box due to their extensive training data, but lack the low-level control necessary to maximize performance in a dense reward embodied environment (which RL excels at).

## 4.4 HOW DOES HYPOTHETICAL MINDS SCALE TO ENVIRONMENTS WITH LARGER POPULATIONS OF AGENTS?

**Running With Scissors in the Matrix Arena (RWS Arena)** is an eight-player extension of RWS, where the agent controls one player against a population of 7 strategies. This adds additional complexity to the decision-making process and tests the scalability of models, as agents now must infer the strategies of separate agents and maintain this information in memory. Hypothetical Minds is set up for this by maintaining separate hypothesis evaluation streams for every agent. Additionally, in order to maximize reward, agents should only interact with opponents that it knows it can beat with its current inventory. Models are therefore tasked with integrating uncertainty and seeking out opponents for which they have high confidence about their strategy.

Hypothetical Minds achieves higher rewards than the baselines in RWS Arena (Figure 2) and is the best model on 6/8 scenarios (Appendix Table 1). HM performs particularly well on homogeneous populations of rock, paper, or scissors (scenarios 5-7). Scenarios 2-4 consist of heterogeneous populations where 2/3rds of the population are one pure strategy and the remaining 1/3rd represents the pure strategy that would beat the best response to the majority strategy (e.g., scenario 2 is 2/3 rock and 1/3 scissors). HM performs the best on 2 out of 3 of these difficult scenarios, and performance for scenario 4 is dragged down by one highly negative episode in which the agent was exploited by the minority strategy. In contrast, all baseline models struggle to achieve rewards of more than 10 on average in nearly every scenario. This highlights the difficulty of the environment, for which proper coordination between high-level plans, the embodiment, and memory is crucial for success. These results also suggest that Hypothetical Minds scales well to population-based environments in which you need to either handle distinct agents differently or make population-level inferences.

## 4.5 ABLATIONS

Our ablation studies focused on three key aspects: (1) comparing performance across different base LLMs, (2) analyzing the components of the ToM module, and (3) analyzing the sensitivity of the ToM module's hyperparameters.

First, we compared GPT-4, GPT-3.5, and Llama-3-70B-Instruct across environments (Figure 5). GPT-4 significantly outperformed the other LLMs. Llama-3 performed moderately on RWS Repeated and Collaborative Cooking, placing between GPT-4 and GPT-3.5, but fell below GPT-3.5 on Prisoner's Dilemma. Due to context length limits, we could not evaluate Llama-3 on RWS Arena.

Second, we conducted a detailed ablation analysis of the ToM module using RWS Repeated as our test environment. We evaluated five variants: 1. **Vanilla Mind Prompting (VMP)**: A simplified version using a single API call for all ToM steps, without hypothesis evaluation or refinement. 2. **Modular Mind Prompting (MMP)**: Removes hypothesis evaluation and refinement but queries the LLM separately for each reasoning step (ie. hypothesis generation and planning the next strategy are separate). 3. **Modular Mind Prompting + Hypothesis Evaluation (MMP + HE)**: This version introduces hypothesis evaluation in comparison to the previous model. 4. **Hypothesis Evaluation and Refinement w/o Mind Prompting (HE + HR)**: Focuses only on hypothesis evaluation and refinement, removing ToM-specific prompting. 5. **Full Hypothetical Minds**: The complete model combining MMP + HE with hypothesis refinement.

Results show MMP consistently outperformed VMP (Figure 4, Figure 8). Adding hypothesis evaluation (MMP + HE) and refinement (full HM) improved performance, with both variants achieving positive returns above 10 across all scenarios. The full model showed particular strength against the challenging best response bot (SC 1). This demonstrates that valuation and refinement may be most beneficial for complex strategies that are difficult to generate or reason about from scratch.

While GPT-4 could sometimes identify correct hypotheses through reasoning alone, it did so inconsistently, highlighting the value of systematic hypothesis evaluation. The HE + HR variant excelled with adaptive opponents but struggled against fixed strategies, suggesting that explicit theory of mind modeling is crucial for recognizing static opponent behaviors (Rakoczy, 2022). Additionally, removing ToM prompting led to verbose, less focused hypotheses (see Figure 9).

Third, we examined key hyperparameters in the ToM module. The 'top_k' parameter controlling how many top hypotheses to evaluate (default=5) trades off computational costs. Performance improves most dramatically when maintaining any hypotheses beyond just the last generated one ('top_k = 0' is equivalent to the MMP model), particularly for challenging scenarios like scenario 1 (Figure 11, Figure 12). This reflects a categorical difference when the best hypothesis is kept around, giving the agent some memory of past guesses, while poor hypotheses naturally fall out of the top 'k'. Increasing 'top_k' beyond 3 shows diminishing returns for RWS Repeated and may become detrimental past 5. The relationship between 'top_k' and computational cost is plotted in Figure 13 and Figure 14. Once top_k > 0, most cost variance stems from other factors, and higher costs ($> \$10$ per episode) actually correlate with decreased performance. This occurs primarily because hypothesis validation reduces subsequent LLM calls, leading to both lower costs and better performance. Tests with Llama3 showed performance is relatively robust to learning rate $\alpha$ (which weights recent predictions) and the validation threshold (Figure 15, Figure 16). While smaller learning rates performed better, the model achieved above-chance performance across all parameter values tested. This robustness makes sense: a good hypothesis will eventually be validated across a reasonable range of these parameters, leading to high rewards for the remainder of the episode.

## 5 CONCLUSION

We have presented Hypothetical Minds, a model that demonstrates strong performance across diverse multi-agent environments through its novel approach to theory of mind modeling and hypothesis evaluation. Our work has several limitations that suggest promising directions for future research. The current implementation requires human intervention to establish scaffolding and prompting for the agent. Additionally, knowledge of game rules and mechanics must be explicitly encoded in the prompts rather than learned from the environment. An important avenue for future research is developing methods for agents to learn these concepts and appropriate types of scaffolding autonomously from environmental feedback. This could enable more general-purpose agents that can flexibly adapt to novel multi-agent scenarios without manual configuration.

The success of our approach in structured game environments suggests broader applications to real-world multi-agent systems where understanding and adapting to people is crucial. Future work should investigate how the principles of the ToM module demonstrated here could extend to more complex social interactions such as adaptive tutoring or personalizing chat bot responses to users with hidden preferences. Such advances would represent important progress toward autonomous agents that can effectively navigate diverse social tasks.

AUTHOR CONTRIBUTIONS

L.C. conceptualized and designed the Hypothetical Minds framework, developed the core codebase, conducted experiments, analyzed results, and wrote the manuscript. V.X. contributed to the framework design, codebase, and implementation. A.B. assisted with experiments. D.Y. and N.H. supervised the research, provided guidance on the project direction, and secured funding. All authors reviewed and provided feedback on the manuscript.

ACKNOWLEDGMENTS

We thank members of the Yamins and Haber labs at Stanford University for their valuable feedback and suggestions throughout the development of this work. This work was supported by the following awards: To D.L.K.Y.: Simons Foundation grant 543061, National Science Foundation CAREER grant 1844724, National Science Foundation Grant NCS-FR 2123963, Office of Naval Research grant S5122, ONR MURI 00010802, ONR MURI S5847, and ONR MURI 1141386 - 493027.

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

# A  APPENDIX

| Scenario | Agent Type | | | | | |
|---|---|---|---|---|---|---|
| | HM (ours) | Reflexion | ReAct | PlanReAct | PPO | OPRE |
| **Running With Scissors Repeated** | | | | | | |
| 0: Mix of all 3 | **50.8 ± 8.6** | 21.0 ± 10.2 | -1.0 ± 3.9 | 5.5 ± 3.6 | -2.9 ± 1.8 | 0.0 ± 0.0 |
| 1: Best Response | **23.2 ± 4.7** | -3.5 ± 3.5 | -2.3 ± 4.2 | 21.1 ± 5.0 | 0.5 ± 1.6 | -0.1 ± 0.0 |
| 2: 0 ∪ 1 | **54.6 ± 4.3** | 29.3 ± 12.5 | 17.1 ± 7.2 | 16.6 ± 6.1 | -1.1 ± 2.0 | 0.0 ± 0.0 |
| 3: Flip After 2 | **40.6 ± 11.1** | 30.9 ± 5.4 | 12.8 ± 4.7 | 13.2 ± 6.4 | 0.8 ± 0.5 | 0.0 ± 0.0 |
| 4: Flip After 1 | **48.5 ± 6.3** | 1.9 ± 4.2 | -1.5 ± 6.1 | 9.0 ± 5.1 | -2.4 ± 1.8 | 0.0 ± 0.0 |
| 5: Gullible | **12.9 ± 5.1** | 8.2 ± 3.2 | 2.5 ± 1.9 | 7.2 ± 3.4 | 0.1 ± 0.6 | 0.1 ± 0.0 |
| 6: Rock | **50.5 ± 4.4** | 17.9 ± 9.5 | 14.3 ± 8.6 | 15.6 ± 4.3 | 1.2 ± 2.0 | 0.2 ± 0.0 |
| 7: Paper | **59.6 ± 4.0** | 13.2 ± 6.8 | 2.7 ± 2.3 | 13.0 ± 4.4 | -0.1 ± 1.0 | 0.1 ± 0.0 |
| 8: Scissors | **63.8 ± 5.1** | 31.0 ± 6.5 | 29.0 ± 4.3 | 14.4 ± 3.3 | -4.2 ± 1.4 | -0.3 ± 0.0 |
| **Running With Scissors Arena** | | | | | | |
| 0: Mix of all 3 | **13.3 ± 6.9** | 1.1 ± 3.6 | 3.5 ± 3.2 | -8.0 ± 4.1 | -0.5 ± 1.5 | 0.1 ± 0.2 |
| 1: Gullible | -2.6 ± 3.4 | **-1.8 ± 2.3** | -5.6 ± 4.0 | 0.9 ± 3.6 | -0.4 ± 1.0 | -0.6 ± 0.2 |
| 2: Rock + Flip After 2 | **16.2 ± 3.9** | -1.7 ± 2.2 | -4.4 ± 2.0 | 2.1 ± 3.2 | -1.0 ± 1.1 | 0.5 ± 0.4 |
| 3: Paper + Flip After 2 | **11.6 ± 4.5** | 1.6 ± 2.2 | -4.2 ± 3.6 | 0.9 ± 3.6 | 1.3 ± 1.4 | 0.3 ± 0.3 |
| 4: Scissors + Flip After 2 | -2.6 ± 5.7 | **10.6 ± 2.2** | -1.2 ± 3.3 | -2.4 ± 3.1 | -2.1 ± 1.0 | 0.2 ± 0.3 |
| 5: Rock | **21.8 ± 3.0** | 1.4 ± 2.0 | -4.3 ± 3.1 | -7.1 ± 2.0 | 1.7 ± 0.8 | 0.5 ± 0.4 |
| 6: Paper | **24.9 ± 2.5** | -1.1 ± 1.4 | -10.7 ± 2.5 | -4.0 ± 3.1 | -2.4 ± 1.0 | 0.4 ± 0.5 |
| 7: Scissors | **32.1 ± 1.8** | 11.8 ± 4.5 | 11.6 ± 2.0 | -0.2 ± 2.6 | 0.9 ± 0.9 | 0.7 ± 0.3 |
| **Collaborative Cooking Asymmetric** | | | | | | |
| 0: Skilled Partner | **628.3 ± 35.1** | 292.6 ± 38.2 | 225.4 ± 60.0 | 402.9 ± 70.1 | 244.0 ± 76.1 | 21.5 ± 15.0 |
| 1: Semi-skilled Partner | **498.8 ± 78.2** | 402.9 ± 65.0 | 254.2 ± 39.1 | 455.6 ± 55.7 | 86.0 ± 28.9 | 36.6 ± 28.0 |
| 2: Unhelpful Partner | **426.8 ± 42.5** | 398.1 ± 33.6 | 383.7 ± 21.4 | 292.6 ± 36.7 | 0.0 ± 0.0 | 62.0 ± 62.0 |
| **Prisoner's Dilemma in the Matrix** | | | | | | |
| 0: 1 ∪ 2 | 40.6 ± 12.9 | 44.5 ± 13.2 | 57.9 ± 13.0 | 37.0 ± 15.3 | 20.7 ± 5.7 | **67.2 ± 1.2** |
| 1: Cooperator | 78.7 ± 2.4 | 87.2 ± 1.7 | 77.3 ± 1.0 | 78.8 ± 2.2 | 40.0 ± 5.4 | **102.5 ± 1.0** |
| 2: Defector | 21.9 ± 0.5 | 22.1 ± 0.8 | 24.0 ± 0.3 | 22.5 ± 0.5 | 9.1 ± 2.0 | **34.5 ± 0.9** |
| 3: Grim Reciprocator | 31.6 ± 4.7 | 28.8 ± 0.4 | 29.2 ± 1.3 | 25.2 ± 0.4 | 13.0 ± 1.1 | **36.3 ± 1.0** |
| 4: Grim Reciprocator (2 strikes) | 27.8 ± 1.2 | 29.3 ± 1.7 | 30.7 ± 1.7 | 28.0 ± 1.2 | 16.5 ± 1.4 | **38.1 ± 1.2** |
| 5: Tit-for-tat | **52.5 ± 1.1** | 33.9 ± 0.2 | 35.8 ± 0.7 | 42.2 ± 2.1 | 15.4 ± 2.1 | 37.4 ± 0.5 |
| 6: Noisy tit-for-tat | **46.7 ± 2.2** | 40.3 ± 2.4 | 41.9 ± 2.7 | 42.0 ± 1.3 | 17.3 ± 2.7 | 47.8 ± 0.6 |
| 7: Cooperates then defects | 20.9 ± 0.5 | 25.1 ± 1.0 | 28.4 ± 0.7 | 24.9 ± 0.6 | 8.8 ± 0.9 | **38.3 ± 0.9** |
| 8: Defector that turns tit-for-tat | **42.1 ± 1.5** | 31.0 ± 1.4 | 30.1 ± 1.4 | 36.8 ± 0.9 | 8.1 ± 2.0 | 38.7 ± 0.6 |
| 9: Defector that turns to noisy tit-for-tat | 45.3 ± 1.8 | 34.7 ± 1.7 | 36.6 ± 1.6 | 40.6 ± 1.1 | 15.1 ± 2.5 | **46.7 ± 0.9** |

Table 1: Average reward and SEM for different agents across substrates and scenarios. Note that the variance in scenarios that are a union of two scenarios may be related to which scenarios were sampled.

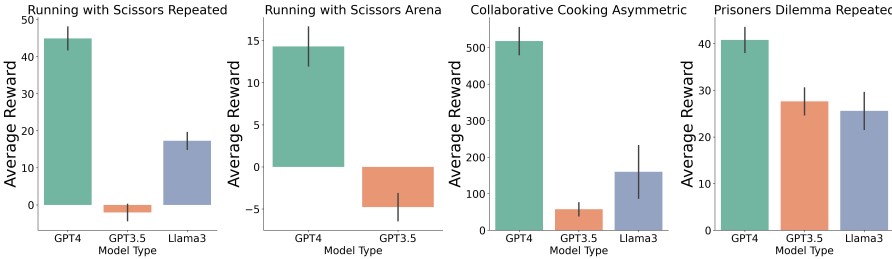

Figure 5: Comparing base LLM models. 3 seeds are generated per LLM.

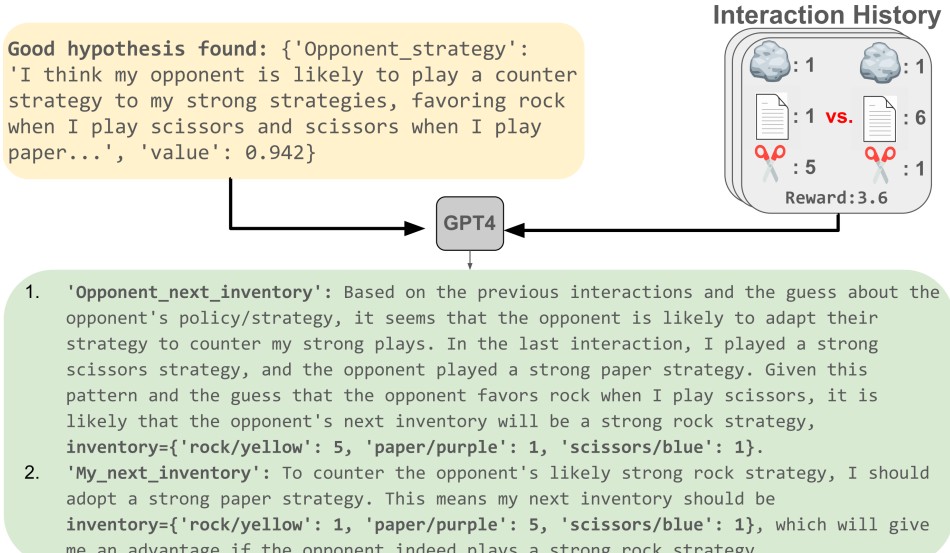

Figure 6: Example of GPT4-based agent finding a good hypothesis and employing good reasoning to select its next target inventory.

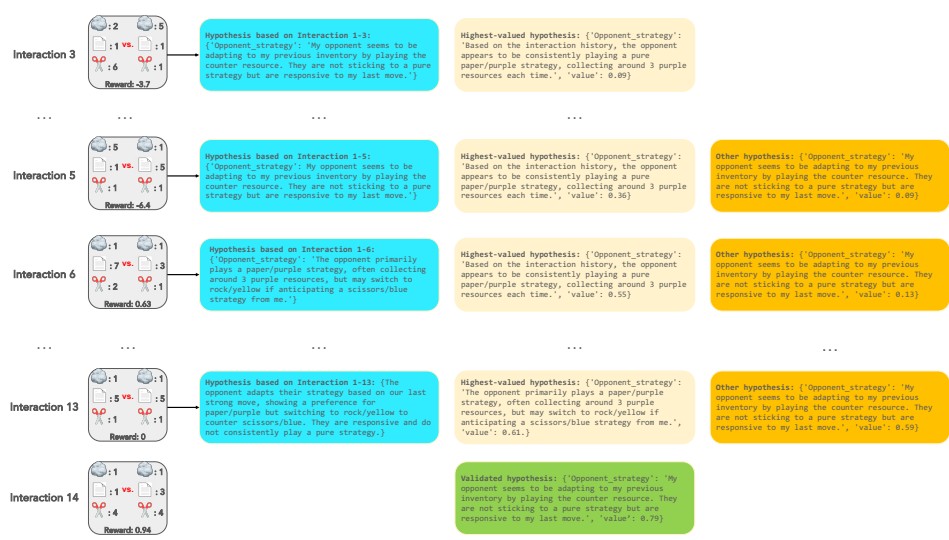

Figure 7: Example of a successful evolution of a hypothesis for MMP + HE + HR playing a Best Response bot. It shows the process of how the LLM-based agent generated a good hypothesis about the opponent's strategy, which was eventually validated.

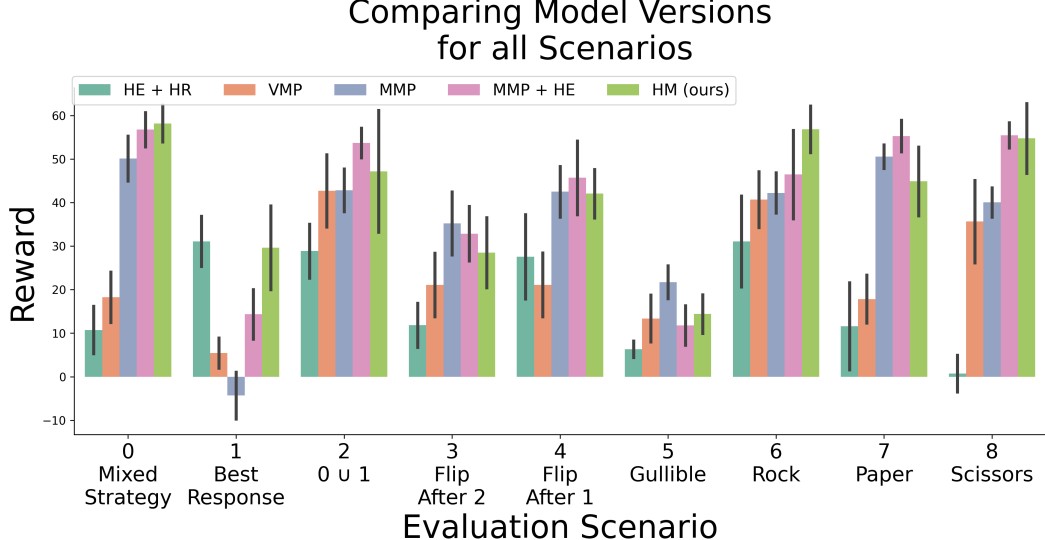

Figure 8: Average reward per episode on all evaluation scenarios for each version of the ToM module using the full version. Errorbars reflect SEM across 5 episodes. Reward is normalized per 1200 timesteps to reduce variance with variable length episodes.

---

**Example generated strategy for HE + HR playing rock bot**

```
Given the interaction history and the current state of the game, it seems that a
strategy that adapts to the opponent's recent choices while maintaining an element of
unpredictability could be effective.  The opponent's strategy is not entirely clear
from the interaction history, but there are patterns of losses and wins that can be
used to inform our strategy.

Here's a high-level strategy for player_0:
1.  Observe the opponent's recent strategy:  Look at the last few interactions
to determine if the opponent is favoring a particular resource.  If a pattern is
detected, prepare to counter it.
2.  Adapt based on interaction history:  If we win an interaction, introduce a slight
randomness to our next choice to avoid being predictable.  If we lose, counter the
opponent's last winning resource.  In the case of a tie, either maintain the current
resource or switch based on the observed pattern of the opponent.
3.  Efficient resource collection:  Focus on collecting resources that align with our
strategy while avoiding unnecessary collection of other resources.  This will help in
building a strong inventory for the next duel.
4.  Strategic duel initiation:  Initiate duels when we have a strong inventory that
counters the opponent's likely choice.  Use memory and observation to infer the
opponent's strategy and position.
5.  Avoid unnecessary resource collection:  Once we have enough of a particular
resource to play a strong strategy, avoid collecting more of that resource and instead
focus on positioning for a duel or collecting a different resource if our strategy
requires it.
6.  Maintain flexibility:  Be ready to adapt our strategy based on new information
from the opponent's actions and the outcomes of interactions.

Now, let's summarize this strategy in a Python dictionary format:
```python
{
'my_strategy': 'Adaptive Strategy with Randomness and Countering',
'observe_recent_opponent_strategy':  True, 'adapt_based_on_interaction_history':
True, 'introduce_randomness_post_win':  True, 'efficient_resource_collection':  True,
'strategic_duel_initiation':  True, 'avoid_unnecessary_resource_collection':  True,
'maintain_flexibility':  True, 'interaction_guidelines':  { 'if_win_next':  'introduce
slight randomness to avoid predictability', 'if_lose_next':  'counter opponent's last
winning resource', 'if_tie_next':  'maintain or switch based on observed opponent
pattern'
}
} ```
```

Figure 9:  Example of a failure mode for HE + HR playing rock bot.  Generated strategies make little sense given the interaction history and requested format.

---

**Example of a successful embodied exploration at the beginning of an episode**

```
, First subgoals:  Subgoal Plan:  Since I am currently at (21, 10) with an orientation
facing North and my high-level strategy is to collect 5 yellow resources to play a
pure rock strategy, I need to find and move towards the nearest yellow box locations.
However, there are no observable yellow box locations in my immediate vicinity.
Therefore, my first subgoal is to explore the environment to find yellow boxes.  I
should move towards the center of the map where I am more likely to find resources and
encounter the other player for a potential duel.

Given that I am at the edge of the map, I should move left (West) towards the center.
I will move to (20, 10), then to (19, 10), and continue in this direction until I find
yellow boxes or reach a more central location on the map.  If I encounter yellow boxes
along the way, I will adjust my path to collect them.  Once I have collected 5 yellow
boxes, I will look for the opponent to initiate a duel.
Action Plan:
```python
{
action_plan:  [ move_to((21, 10), (20, 10)), move_to((20, 10), (19, 10)), move_to((19,
10), (18, 10)), move_to((18, 10), (17, 10)), move_to((17, 10), (16, 10))]
} ```
```

Figure 10: Example of a subgoal module exploring the environment at the beginning of an episode to build up a cognitive map of where resources are located.

# B   HYPERPARAMETERS AND COMPUTATIONAL COSTS

| Hyperparameter | Value |
|---|---|
| Alpha ($\alpha$) | 0.3 |
| Intrinsic reward c | 1.0 |
| Threshold for Hypothesis Value ($V_{thr}$) | 0.7 |
| Top $k$ Hypotheses Considered ($k$) | 5 |

Table 2: Default Hyperparameters of the Hypothetical Minds model

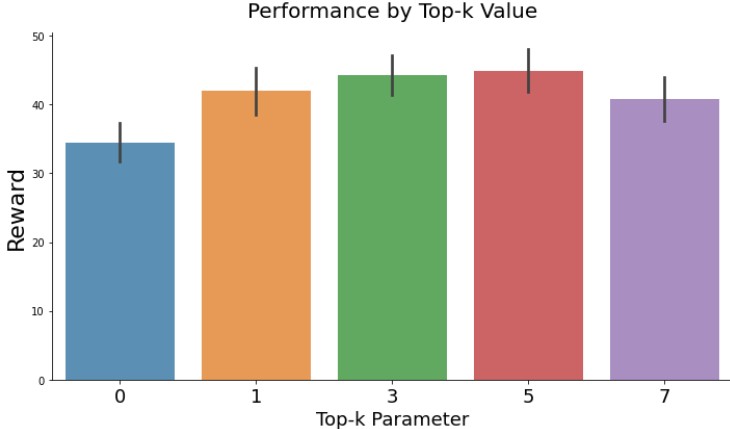

Figure 11: Average reward per episode on Running With Scissors Repeated (all scenarios) for four values of top_k. Top_k reflects the number of top hypotheses to continue evaluating (default=5). Errorbars reflect SEM across 45 episodes. Reward is normalized per 1200 timesteps to reduce variance with variable length episodes.

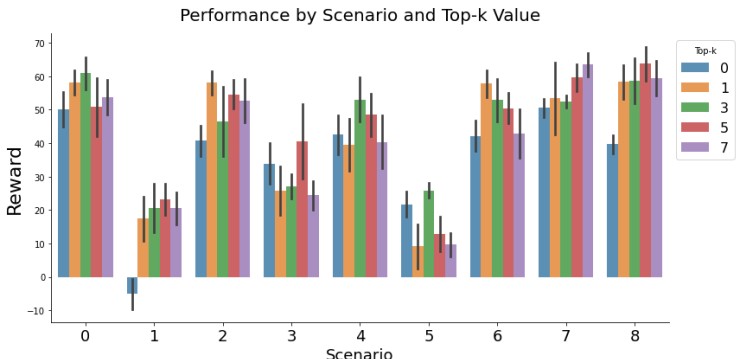

Figure 12: Average reward per scenario on Running With Scissors Repeated for four values of top_k. Top_k reflects the number of top hypotheses to continue evaluating (default=5). Errorbars reflect SEM across 5 episodes per scenario. Reward is normalized per 1200 timesteps to reduce variance with variable length episodes.

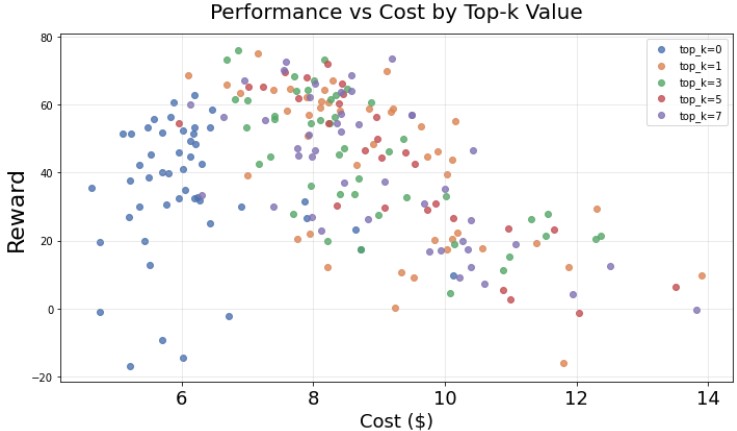

Figure 13: Scatter plot showing the relationship between computational cost and reward for different top-k parameter values in the hypothesis generation process. Each point represents an episode, with color indicating the top-k value used. Reward and cost is normalized per 1200 timesteps to reduce variance with variable length episodes.

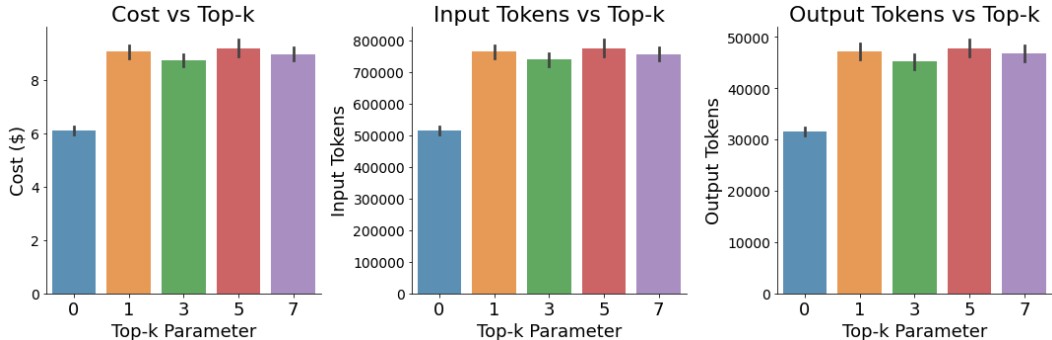

Figure 14: Comparison of computational costs across different top-k values in three metrics. Left: Total monetary cost in dollars. Middle: Number of input tokens consumed. Right: Number of output tokens generated. All metrics are normalized per episode length (1200 steps). Errorbars reflect SEM across 45 episodes.

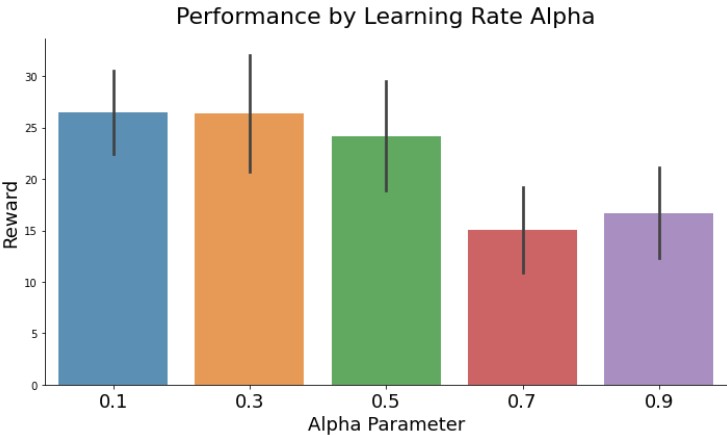

Figure 15: Average reward per episode on Running With Scissors Repeated (all scenarios) for each four values of the learning rate alpha $\alpha$. Alpha reflects how much to weigh recent interactions when evaluating the prediction accuracy of the hypotheses, with higher values weighing recent interactions more heavily (default=0.3). Errorbars reflect SEM across 10 episodes in two representative scenarios. Reward is normalized per 1200 timesteps to reduce variance with variable length episodes.

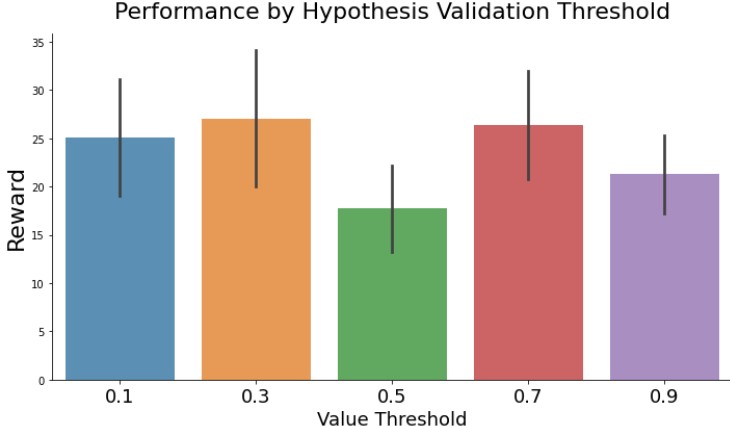

Figure 16: Average reward per episode on Running With Scissors Repeated (all scenarios) for each five values of the value threshold for hypothesis validation. V_thr reflects how confident we want to be before validating a hypothesis as correct such that we do not generate or evaluate more hypotheses until that value falls under the threshold (default=0.7). Errorbars reflect SEM across 10 episodes in two representative scenarios. Reward is normalized per 1200 timesteps to reduce variance with variable length episodes.

| Hyperparameter | Value |
|---|---|
| Alpha ($\alpha$) | 0.3 |
| Counterfactual reward c | 3.0 |
| Threshold for Hypothesis Value ($V_{\text{thr}}$) | 3.0 |
| Top $k$ Hypotheses Considered ($k$) | 5 |

Table 3: Hyperparameters of the HE + HR model

| Hyperparameter | GPT-4 | GPT-3.5 | Llama3 |
|---|---|---|---|
| Model | "gpt-4-1106-preview" | "gpt-3.5-turbo-1106" | "Meta-Llama-3-70B-Instruct" |
| Max tokens | 4000 | 2000 | 2000 |
| Temperature | 0.1 | 0.2 | 0.2 |
| Top p | 1.0 | 1.0 | 1.0 |
| n | 1 | 1 | 10 |

Table 4: Hyperparameters of Various Models

## B.1 EXPERIMENTS COMPUTE RESOURCES

In Table 5, we show the cost of running an episode with 1000 steps. For Llama3 experiments which required 4 GPUs, we ran our experiments on three compute nodes with A40s or L40s.

| | Money ($) | Time (mins) |
|---|---|---|
| **HM-GPT4** | 10 | 45 |
| **Reflexion** | 8 | 42 |
| **ReAct** | 6 | 14 |
| **PlanReAct** | 4 | 21 |
| **HM-GPT3.5** | 1 | 24 |
| **HM-Llama3** | - | 58 |

Table 5: Experiment costs for a single episode of RWS Repeated. Rounded to nearest integer.

## C  Environments

### C.1  Running With Scissors Repeated

Specifically, here we evaluate our model on the *Running With Scissors in the matrix: Repeated* environment (RWS) in the Melting Pot multi-agent decision-making benchmark (Agapiou et al., 2022). This is a zero-sum competitive environment with two players moving around a map and collecting yellow, purple, or blue resources that correspond to rock, paper, and scissors respectively. In addition to movement, the agents have an action to fire an "interaction" beam which initiates a duel with the other player when that player is within range of the beam. An interaction results in one agent getting positive reward and the other agent getting an opposite negative reward according to the inventories of resources picked up by each player. Specifically a player will collect an inventory, which is only observable by that player:

$$\rho = (\rho_{yellow}, \rho_{purple}, \rho_{blue}).$$

Reward is determined by matrix multiplication operations mirroring the rock, paper, scissors matrix game:

$$r_{\text{row}} = \mathbf{v}_{\text{row}}^T A_{\text{row}} \mathbf{v}_{\text{col}}, \qquad\qquad r_{\text{col}} = -r_{\text{row}}$$

where $v_i = \frac{\rho_i}{\sum_{j=1}^{K} \rho_j}$ and

$$A_{\text{row}} = \begin{bmatrix} 0 & -10 & +10 \\ +10 & 0 & -10 \\ -10 & +10 & 0 \end{bmatrix}.$$

The partially-observable input in Melting Pot consists of a 5x5 window around the agent such that it can see three grids in front of itself and one behind it, and two on each side.

### C.1.1  Scenarios

Description of scenarios for each substrate are reproduced directly from (Agapiou et al., 2022):

**SC0:** *Versus mixed strategy opponent.* Here the focal agent must defeat an opponent that was trained to play a pure strategy: either rock, paper, or scissors. However, the specific opponent is sampled at test time so it could be any of those. All opponents commit strongly to their choice, aiming to collect at least three resources before interacting. To defeat them, the focal agent should scout out which pure strategy its opponent is playing and then collect the resources to implement its counter strategy. Since this is a one-shot interaction, success requires the focal agent to pay close attention to which resources are missing since they provide a clue to which strategy their opponent is implementing.

**SC1:** *Versus opponent who plays the best response to what the focal player did in the last round.* Here the focal agent must defeat an opponent who may change their strategy with each interaction. The opponent will always select the best response to what the focal player selected in the previous interaction. For instance, if the focal player plays rock in one interaction then its opponent will play paper in the next interaction. On the first interaction of each episode it chooses one of the three pure strategies at random. The opponent always commits strongly to its choice, aiming to collect at least five resources before interacting. To win the focal player can trick its opponent into choosing a specific strategy and countering it. This requires changing strategy from interaction to interaction, cycling around the three options.

**SC2:** *Versus opponent who sometimes plays a pure strategy but sometimes plays the best response to what the focal player did in the last round.* Focal player must defeat an opponent sampled from the union of the background populations used in **SC 0** and **SC 1**. The probability of sampling a pure opponent is 3/4 while the probability of sampling a best response opponent is 1/4.

**SC3:** *Versus mixture of opponents who often flip to other strategies after two interactions.* Focal player must defeat an opponent that may initially play any pure strategy and, with probability 1/3, may flip after the second interaction to the best response to the best response to

its initial strategy. For example if it starts out playing rock then the best response to that would be paper, so after the second interaction it would switch to playing the best response to paper i.e. scissors. It only weakly commits to its strategy for the first two interactions. That is, it aims to collect only one resource before interacting. After two interactions, at the point when it changes strategy, it also starts committing more strongly to its choice, aiming to collect five resources before each interaction. With probability 2/3, the opponent instead plays a pure strategy throughout the entire episode. In half of the pure opponent episodes the bot fully commits to its pure strategy, aiming to collect five resources before interacting, while the other half of the time it commits less strongly, aiming to collect only one resource before interacting. Note that opponents may be weakly committed to their strategy for the first two interactions regardless of whether they will ultimately flip strategy or not so it's not possible for the focal agent to observe weak commitment early on as a cue to predict whether or not their opponent will later flip strategies.

**SC4:** *Versus mixture of opponents who either flip to another strategy after one interaction and keep it forever or continue to change, always best responding to what the focal player just did.* Two kinds of opponents are possible. Both change their strategy after the first interaction. With probability 3/4 the opponent will be a bot that flips to a different strategy after the first interaction and then follows it till the end of the episode. It always flips to the best response to the best response to its initial strategy (so if it initially plays rock then it will flip to scissors). With probability 1/4 the other kind of opponent is sampled. This opponent is identical to the one in SC 1. Both kinds of opponents always fully commit to their choice, aiming to collect at least five resources before interacting so its not possible to observe the opponent's commitment level to predict which kind they are. To win the focal player must figure out which kind of opponent it is playing against and either best respond by selecting the same choice in all interactions after the first if paired with the first kind of opponent, or apply the cyclic strategy described as the solution to SC 1 if paired with the second kind of opponent.

**SC5:** *Versus gullible opponent.* Here the focal agent must defeat an opposing agent that was trained to best respond to agents playing pure strategies. The opponent should attempt to scout out what strategy the focal agent is playing so it can pick the appropriate counter. To defeat it, the focal agent should feint toward one resource and then collect the counter to its counter. So for example, if the focal agent successfully feinted that it would pick rock, inducing its opponent to pick paper, the focal agent should then collect and play scissors. This opponent is fairly weak.

**SC6:** *Versus pure rock opponent.* Opponent always plays rock, and commits to it strongly, aiming to collect five resources before interacting. The focal player gets a high score when it picks paper and commits strongly to that choice.

**SC7:** *Versus pure paper opponent.* Same as SC 6 but opponent plays paper so focal player should play scissors.

**SC8:** *Versus pure scissors opponent.* Same as SC 6 but opponent plays scissors so focal player should play rock."

It should be noted that scenario 5 includes a gullible opponent, that attempts to scout out what strategy you are playing and pick the appropriate counter. From first principles, to beat this opponent an agent should feint towards one resource to fool the opponent and then select the counter to its counter. Since our ToM module selects strategies on a higher-level of abstraction than the embodiment, it was not specifically designed to beat this opponent. Integrating high-level strategy information with relevant embodied information is the subject of future research. However, in practice the gullible bot does not effectively scout out our agent, and it frequently plays the same strategy, which is exploited by our agent, leading to positive rewards on this scenario on average for all model versions.

## C.2 RUNNING WITH SCISSORS ARENA

We also evaluate our model on *Running With Scissors in the matrix: Arena* environment in the Melting Pot multi-agent decision-making benchmark (Agapiou et al., 2022). This environment has the same dynamics as *Running With Scissors in the matrix: Repeated* with the main exception being that

their are $8$ players in this substrate playing on a larger $25$ by $24$ matrix with a $11$ by $11$ observability window, skewed towards viewing more in front of the agent. All scenarios under this agent represent one focal resident (agent) playing against 7 others with varying fixed and dynamic policies, attempting to maximize reward decided by the payoff matrix equivalent to the one in Running with Scissors Repeated.

### C.2.1 SCENARIOS

**SC0:** *Versus a background population containing bots implementing all three pure strategies.* Here one focal player joins seven from the background population. The background population contains bots who implement all three pure strategies: rock, paper, and scissors. They may either commit to their strategy moderately (aiming to collect three resources before interacting) or more strongly (aiming to collect five). The task for the focal agent is to watch its opponents, see what strategy one of them is implementing, and act accordingly.

**SC1:** *Versus gullible bots.* Here one focal player joins seven from the background population. The background population consists entirely of weak bots who were trained to best respond to agents playing pure strategies. They are weak opponents.

**SC2:** *Versus mixture of opponents who play rock and some who flip to scissors after two interactions* Here one focal player joins seven from the background population. The focal player should pay attention to what each prospective partner has collected since 2/3 of them play rock while 1/3 play scissors after the first two interactions. Choosing paper to best respond to rock is a bad choice if accidentally paired with an opponent playing scissors.

**SC3:** *Versus mixture of opponents who play paper and some who flip to rock after two interactions.* Like SC2 but with bots playing paper and bots switching from paper to rock.

**SC4:** *Versus mixture of opponents who play scissors and some who flip to paper after two interactions.* Like SC 2 but with bots playing scissors and bots switching from scissors to paper.

**SC5:** *Visiting a population of pure paper bots.* Here one focal player joins seven from the background population. All seven background bots play paper so the focal player can get a high score by playing scissors.

**SC6:** *Visiting a population of pure rock bots* Here one focal player joins seven from the background population. All seven background bots play rock so the focal player can get a high score by playing paper.

**SC7:** *Visiting a population of pure scissors bots. Here one focal player joins seven from the background population.* All seven background bots play scissors so the focal player can get a high score by playing rock.

### C.3 PRISONERS DILEMMA REPEATED

The *Prisoners Dilemma in the Matrix Repeated* environment is a mixed-motive one where two individuals collect resources that represent 'defect' (red) or 'cooperate' (green) and compare inventories in an encounter, analogous to the Running With Scissors substrates. Consequences of the inventory comparison are congruent with the classic Prisoner's Dilemma matrix game, exposing tension between reward for the group and reward for the individual. Reward is delivered after an interaction weighted by the proportion of items in each agent's inventory, as described above for Running With Scissors. The payoff matrix for the interaction is

$$A_{\text{row}} = A_{\text{col}}^T = \begin{bmatrix} 3 & 0 \\ 5 & 1 \end{bmatrix}$$

### C.3.1 SCENARIOS

**SC0:** *Partner may play either cooperate or defect* The optimal strategy is simply to unconditionally defect. However, given that the focal doesn't know the strategy of the background player, a good strategy is more subtle. A reasonable strategy is to be a grim reciprocator cooperator, which would cooperate with the cooperator, and defect to the defector. Alternatively the focal player might try to ascertain whether the background player is exploitable.

Doing so, however, carries a risk, for if the background player were to be a Grim reciprocator (like in other scenarios), this would cause them to defect for the rest of the episode.

**SC1:** *Partner typically plays cooperate.* The optimal strategy is simply to unconditionally defect. The same considerations about uncertainty of the background player's strategy from Scenario 0 apply here.

**SC2:** *Partner typically plays defect* The optimal strategy is simply to unconditionally defect. However, because the focal player doesn't a priori know the strategy of the background player, they must first try to find out their strategy. This can be done by looking at which resources they collect or by paying attention to the results of the first few interactions. Once the focal has identified its background partner is defecting then it may have confidence that it should defect as well. The focal player should also consider the possibility that the background bot is corrigible, i.e. that it could be persuade to switch from defection to cooperation. This is not the case here but the background populations used in SC 8 and SC are corrigible.

**SC3:** *Partner is a hair-trigger grim reciprocator, i.e. one who initially cooperates but, if defected on once, will retaliate by defecting forever after.* The optimal strategy is simply to cooperate. Grim reciprocator background players are non-exploitable, and there is no way to know how they will react to a defection ahead of time. Because of this uncertainty, testing for exploitability can lead to poor performance of the focal player. Conditional cooperators who cooperate first but retaliate if defected on should achieve a high score.

**SC4:** *Partner is a two-strikes grim reciprocator, i.e. one who initially cooperates, but if defected on twice, will retaliate by defecting forever after* The optimal strategy is simply to cooperate. Grim reciprocator background players are non-exploitable, and there is no way to know how they will react to a defection ahead of time. Because of this uncertainty, testing for exploitability can lead to poor performance of the focal player. In principle, it would be possible to defect once against the background player leading to higher reward. But since it is not possible to know the background player is a two-strikes grim reciprocator, and testing it against a hairtrigger grim reciprocator leads to defection, in practice is better simply to cooperate. Conditional cooperators who cooperate first but retaliate if defected on should achieve a high score.

**SC5:** *Partner is a tit-for-tat conditional cooperator* The optimal strategy is simply to cooperate. Defecting against a tit-for-tat agent, even occasionally, might lead to miscoordinated interactions where one player cooperates and the other defects, in an alternating way. Forgiveness is one way to break out of such cycles of recrimination. Conditional cooperators who cooperate first but retaliate if defected on should also be forgiving to ensure they do well in this scenario.

**SC6:** *Partner is a tit-for-tat conditional cooperator who occasionally plays defect instead of cooperate.* Like the previous scenario, except the tit-for-tat background player occasionally will defect instead of cooperate. This is known as trembling hand in game theory. A strict tit-for-tat focal player would occasionally fall into miscoordinated interactions with the background player resulting in alternating cooperation and defection. As in SC5, focal conditional cooperators must also be forgiving to ensure they do well in this scenario. Forgiveness is even more important here since the background player will defect relatively frequently itself but will still implement tit-for-tat retaliation when defected on itself.

**SC7:** *Partner plays cooperate for a while then switches to defect* Similar considerations to the previous scenarios. A good strategy is a grim reciprocator, or tit-for-tat for the focal player. Unconditional cooperation would be exploited by the background player.

**SC8:** *Partner tries to take advantage of the focal player by playing defect, but if punished, partner then switches to tit-for-tat conditional cooperation* Related to Scenario 2, the optimal strategy is for the focal player to persuade the background player to stop defecting by punishing it through defecting itself. Once persuaded, the background player implements a conditional cooperation (tit-fortat) strategy. So it is safe to start cooperating with them once you have verified that they are themselves consistently cooperating.

**SC9:** *Partner tries to take advantage of the focal player by playing defect, but if punished, partner then switches to noisy tit-for-tat conditional cooperation* Like the previous scenario, except

the focal player must implement a more generous form of conditional cooperation after persuading the background player to switch from defection.

## C.4 COLLABORATIVE COOKING ASYMMETRIC

In the *Collaborative Cooking Asymmetric* substrate, players need to collaborate to follow recipes. The environment described in (Agapiou et al., 2022) follows the regular pseudoreward scheme, which is turned off by default. The asymmetric environment is a version of the Collaborative Cooking with an asymmetric advantages map. This is to test whether players can choose high-level strategies that play to their strengths.

### C.4.1 SCENARIOS

**SC0:** *Collaborate with a skilled chef* Here the background player implements a particular policy that can be very effective when its partner does its part. The two players are on two distinct and disconnected sides of the map. On one side the goal delivery location is close to cooking pots and the tomato dispenser is far away whereas on the other side the the goal delivery location is far from the cooking pots but the tomato dispenser is close. The players should collaborate, each specializing in the part of the task that it is most efficient for them to do on the side of the map where they spawned. The background player implements this kind of policy, which depends on the actions of its partner to complete the task. The background player was trained with the V-MPO algorithm.

**SC1:** *Collaborate with a semi-skilled apprentice chef* This scenario is similar to SC 0 but the background player is not as well trained. In fact the background population used here is the same as in SC0 but from an earlier point in training. The importance of evaluating cooperation with bots of varying skill levels, and different points in training.

**SC2:** *Succeed despite an unhelpful partner* In this scenario the background player never moves or helps in any way. On this map it is less efficient to implement all steps of the recipe alone versus to work together with a partner. But it is still possible for either player to perform all the steps on their own. The task is to realize that the background player won't do their part of the joint policy so the focal agent had better do everything itself.

# D  METHODS

## D.1  TEXTUAL MAP

Pixel images of the global state are preprocessed into a text-based state representation. The images are divided into 8x8 patches, each corresponding to a cell within the Melting Pot grid that entities can occupy. Each patch may represent one of four types: an agent, a resource, a wall, or a blank space. The patches are labeled by comparing them to manually labeled reference patches of each type of entity, including the numerous possible body orientations and hat colors an agent can embody in each environment. Automating the process with computer vision is a candidate for future work. The global state can then be fully represented in text by coordinates in a Height x Width grid and the entity label at each coordinate. Egocentric states are then created from this according to the partially-observable 5x5 box around an agent (dependent on its orientation). We tried several representations of feeding this textual map to GPT, and in practice the best representation consists of printing each entity type with a list of all the coordinates where that entity type is present. For example "Player Position: {'player_0-S': [(21, 4)]}, Observable Yellow Box Locations: [(13, 10), (14, 11)]}, Observable Blue Box Locations: [], Observable Purple Box Locations: [(13, 11), (15, 11)]" encodes the player position and orientation (south) along with the observable box locations. The player's current inventory is also included in the textual state representation, as it is in the default state representation for Running With Scissors and Prisoner's Dilemma.

## D.2  MEMORY

The memory system consists of two parts. The first data structure appends the observed states in the previous step to lists of each entity type in a tuple with the step it was observed. For example: 'yellow_box': [((13, 3), 'Step: 1087'), ((13, 4), 'Step: 1087'), ((7, 3), 'Step: 1091')]. The LLM is

prompted that its memory can be outdated and therefore should take the step it was last observed into account.

The second data structure in the memory system contains a list of the agent's inventories and rewards from the interactions that occurred so far. This specific information, distinct from the previously observed states, is relayed to the ToM module.

### D.3 THEORY OF MIND MODULE

The Theory of Mind Module is queried periodically after discrete events. For the *∗ in the Matrix* substrates, this occurred after an interaction. For collaborative cooking, this occurred after a dish was delivered.

The ToM module consisted of a multi-step process, as depicted in Figure 1 for the *∗ in the Matrix* substrates:

1. **Record the observed behavior from the other agent's trajectory** $\phi(\tau)$**.** Here this refers to whether they played rock, paper, or scissors, the argmax of the inventory (or cooperate/defect in Prisoner's Dilemma). Since the opponent's inventory is never observed, we have to estimate it given the inventory Hypothetical Minds played and the reward it received. Thus, we ask the LLM to estimate the opponent's inventory given this information, and note the output as the empirical opponent's inventory.

2. **Evaluate Hypotheses about opponent's strategy.** In the previous interaction, the top k hypotheses are used to generate predictions about the opponent's next inventory $\hat{\phi(\tau)}$. These argmax of the inventory predictions are compared to the argmax of the empirical opponent's inventory (did they play rock, paper, or scissors). As described in the main text, Hypotheses that led to correct predictions get a positive intrinsic reward and negative otherwise. If a hypothesis is validated, meeting $V_{\text{thr}}$, then step 3 is skipped and this hypothesis is used for step 4 until the hypothesis falls below the threshold (meaning its not longer making good predictions)

3. **Generate new and refine old hypotheses**. The LLM is tasked with generating a hypothesis about the other agent's strategy given the entire interaction history (see prompt below for more details). The prompt also includes the top k hypotheses generated so far if the hypotheses have a value above 0 (meaning at least one correct prediction) such that the LLM can refine previously generated hypotheses.

4. **Guess opponent's next goal.** The LLM is prompted to guess the opponent's next inventory given a hypothesis and the interaction history. If no hypothesis has yet surpassed $V_{\text{thr}}$, then guesses are made for the top k hypotheses and the last generated hypothesis. The prediction from the last generated hypothesis would be used to select a counter inventory in the next step. Moreover, all the predictions will be used for step 2 (hypothesis evaluation) after the next interaction. If a hypothesis crosses $V_{\text{thr}}$, then only it will be used in this step.

5. **Select goal to counter opponent.** The LLM is prompted to select a counter inventory given the prediction about the opponent's next inventory. Since this step involves straightforward reasoning given the opponent's predicted inventory, it is done simultaneously in the API/LLM call with the previous step. Therefore, the LLM is specifically tasked with outputting both the predicted opponent's next inventory $\hat{\phi(\tau)}$ and its own goal/target inventory $z$ in a single API/LLM call.

In RWS Arena, step 4 is done separately than step 5. The LLM is first prompted to guess the next inventory for the opponent they just played. Then in a subsequent step, it is asked again to select which opponents to seek out, and to guess what inventory they will play (this could be a different opponent than the one interacted with in the last round). Therefore, the ToM module can evaluate the hypotheses for each opponent based on the quality of the predictions they make for that particular opponent, and this process is separable from selecting the target inventory in the next interaction (see prompts).

A similar process occurs for *Collaborative Cooking Asymmetric*. Step 1 is hardcoded and not LLM dependent; the other agent's actions/behavior $\phi(\tau)$ are labeled by custom code given the observations (abstracting away the problem of action recognition from textual observations). These actions

include "Teammate picked up a dish", "Teammate put down a dish", "Teammate picked up cooked soup in dish", "Teammate delivered cooked soup", "Teammate picked up a tomato", or nothing. In step 4, the LLM is prompted to guess the teammate's next behavior $\phi(\hat{\tau})$ in natural language. Another LLM instance is used in step 2 to assess whether the prediction was correct or not, prompting the LLM to output True or False, and giving each hypothesis the appropriate intrinsic reward. Step 5 is completed in a separate API call for Collaborative Cooking, and the LLM is prompted: "what strategy do you want to take next and why? Teammate's observed strategy: ". Think step by step about how to adapt to their behavior and maximize all resources and efficiency accordingly."

### D.4  SUBGOAL MODULE

The subgoal module is responsible for generating efficient subgoal plans for the agent. Given the high-level strategy and the current state of the game, the module decomposes the strategy into a sequence of subgoals in the form of action function calls to efficiently implement the strategy. The subgoal module uses the LLM to generate these plans as a sequence of action function calls (usually 3-6). The prompt includes the current step of the game, the high-level strategy/target inventory previously decided upon, details about the current observations (including player position, orientation, inventory, observable resource locations, other agent locations), valid movement locations, memory, and instructions about the action functions.

Action functions for the * *in the Matrix* substrates:

- move_to(src_coord, target_coord): Efficiently move agent from source coordinate to target coordinate.
- fire_at(target_coord): Stay around specified coordinate and fire interaction when opponent is spotted to initiate duel.

Action functions for *Collaborative Cooking Asymmetric*:

- move_to(src_coord, target_coord): Efficiently move agent from source coordinate to target coordinate. Only move to valid move_to locations where counters or objects are not present.
- interact(target_coord): Move to and interact with the entity at the target coordinate, such as picking up ingredients or delivering dishes of cooked soup. To place an object on a counter to free your hands, use interact(counter_coord).
- wait(target_coord): Wait for the pot at target_coord to finish cooking. Check the progress of the pots and only use valid locations where pots are present.

### D.5  ACTION PLANNER

The action planner turns the sequence of subgoals specified by the subgoal module into a sequence of atomic actions compatible with the Melting Pot environment. These actions include step forward, backward, left, or right, turn left or right, fire zapping beam, and noop. For the `move_to(source_coordinate, target_coordinate)` function, the A* algorithm find the most efficient path given a set of obstacles. Walls are always considered obstacles, and other resources are conditionally added as obstacles. If a path can be found without picking up another resource, the action planner will return it. However, in many cases the language model picks a target coordinate where other resources cannot be avoided. The action planner will use a priority system in this case, privileging paths where only resources of the same type as the target coordinate will be picked up. The `fire_at` function returns a sequence of actions to turn towards the opponent and zap them when the agent is within the extent of the zapping beam's range. If the opponent is not in view, then the fire_at function turns the agent continually clockwise until the other agent is found.

### D.6  SELF-REFLECTION - COLLABORATIVE COOKING

For *Collaborative Cooking Asymmetric*, an evaluator and self-reflection mechanism was added as in the Reflexion baseline (Shinn et al., 2024). This was added such that if the agent was making action plans that did not change the state of the world, for example trying to pick up a tomato while holding a dish, these action plans were not repeated with the same state information. By first reflecting on

whether the previous action plan was successful, the agent was able to make less mistakes over the course of the episode. This additional cognitive module for self-reflection could in principle also be added for the other substrates, but was not necessary for good performance on the * *in the matrix* games because the state transitions were simpler for LLMs to understand.

# E    BASELINES

## E.1    REACT

The ReAct (Yao et al., 2023) agent combines reasoning traces with task-specific actions in an interleaved manner. This approach allows the agent to generate both reasoning steps and actions within the same language model framework. In our context, the reasoning consists of chain of thought reasoning preceding a subgoal plan in the specified format of utilizing action functions. The agent is prompted to think about the other agents' strategies and come up with a subgoal plan accordingly. Thus, ReAct is functionally an ablation of Hypothetical Minds such that there is only a subgoal module and not a theory of mind module. Three example responses are shown as few-shot prompts, consistent with the ReAct framework.

## E.2    REFLEXION

Reflexion adds evaluation and self-reflection to the ReAct agent backbone serving as the Actor module. After a subgoal plan is completed, the LLM is queried to evaluate the outcomes of that plan. Outcomes are represented as the reward during the plan, and salient state information pre and post plan. This state information includes the position and the inventory of the agent for the * *in the matrix* games. For *Collaborative Cooking Asymmetric*, the given state information included position, what the agent is holding, and the state of the two pots.

## E.3    PLANREACT

We include the PlanReAct architecture introduced in (Liu et al., 2023b) as a hierarchical baseline. This model first generates a high-level plan in language and then feeds this plan to a subgoal module that outputs a subgoal plan based on the high-level plan. Thus, the only difference between PlanReAct and Hypothetical Minds is that high-level planning is mediated by the multiple processing steps of the theory of mind module, including hypothesis generation, evaluation, and refinement.

## E.4    PPO

We train RL agents in a population of PPO agents (Schulman et al., 2017) on each substrate. The weights are randomly initialized for each agent in the population and weights are not shared. Therefore the agents are not playing against identical copies of themselves and see a greater diversity during training than traditional self-play. Models were trained in PyTorch using the Ray Rllib pipeline and this starter code https://github.com/rstrivedi/Melting-Pot-Contest-2023. Optimal parameters were searched over and the final models were trained for 1e8 steps.

## E.5    OPRE

We include the Options as REsponses model (OPRE) introduced in the first version of the Running With Scissors environment (Vezhnevets et al., 2020). OPRE is a hierarchical MARL method where agents learn high-level options as strategic responses to other agents' behaviors. The high-level controller selects options based on observations of other agents, and the low-level controller executes actions conditioned on these options. As this was included as a baseline in the Melting Pot 2.0 paper, we reuse the results reported in that paper (Agapiou et al., 2022).

# F    ABLATION DETAILS

The Hypothesis Evaluation + Hypothesis Refinement model had a different evaluation procedure than the other models. Rather than computing values based on predicting the opponent's inventory,

here we use extrinsic reward and counterfactual reward. If a hypothesis is used online for goal selection, then the rewards received in the next interaction can be directly used for evaluating it. For the other considered hypotheses, we simulate counterfactual reward by 1. asking GPT to generate a target inventory given the hypothesis/strategy and the given situation and 2. after the next interaction we ask GPT again to reason about what the reward would have been if it played the inventory from 1. GPT is asked to output, positive, negative, or neutral, which we convert to reward with the $c$ parameter.

## G  PROMPTS

All prompts can be seen in our code. The most representative prompts are also reproduced below.

## G.1 Running With Scissors: Arena Prompts

---

**RWS Arena System Message**

You are Agent 0 in the eight player 'running_with_scissors' Melting Pot multiagent reinforcement learning environment that is a 25x24 (x by y) grid with resources to collect and walls to navigate around. 8 Players can move around the map and collect resources of 3 discrete types corresponding to rock, paper, and scissors strategies – Yellow box = rock – Purple box = paper – Blue box = scissors. Rock/yellow beats scissors/blue, paper/purple beats rock/yellow, and scissors/blue beats paper/purple.
In addition to movement, the agents have an action to fire an "interaction" beam which initiates a duel with one player getting positive reward and the other agent getting an opposite negative reward according to their inventories.
All players carry an inventory with the count of resources picked up since last respawn and for each respawn start with an inventory of 1 resource each. This inventory is visible in the state with the key 'inventory'.
To play a pure strategy strongly, pick up at least 5 resources or more of the color and then fire the interaction beam at another player. To commit less strongly to a strategy, pick up around 3 resources of the color and then fire the interaction beam at another player.
Usually you will only want to pick up one type of resource before an interaction, in order to gain the most information about the other players' strategies and to not waste time collecting other resources.
You also want to maximize the number of interactions so after you pick up 4-6 resources, you should seek out a duel to reset your inventory and gain more information about the other players' strategies.
Your opponents will also almost always only pick up one type of resource before an interaction.
For example, player0_inventory = [7, 1, 1] (Yellow, Purple, Blue) is a good inventory that will lead to an informative duel, whereas player0_inventory = [2, 2, 2] (Yellow, Purple, Blue) will not be informative.
Your reward is the result of a matrix multiplication involving your inventory in a vector format, and your opponent's inventory vector, and a payoff matrix similar to rock paper scissors.
r_t = transpose(your_inventory) * A_payoff * opponent_inventory where A_payoff = np.array([[0, -10, 10], [10, 0, -10], [-10, 10, 0]])
The reward usually ranges from (5, -5) depending on the inventories of both players (the min is -10 and max 10, but it is rare to get these magnitudes). Typically +/- 3-5 is a high magnitude, and a reward near 0 suggests both players played a similar inventory.
State Description: This environment is partially-observable, you can observe an 11x11 grid around your agent depending on your position and orientation (you can see more in front of you than behind).
Previously seen states will be represented in memory, but note that these states could potentially be outdated. For example, the other agent could collect a resource that you previously saw.
Given the partially-observable nature of the environment, you will need to explore the environment appropriately and select goals based on the information you've gathered.
Also pay attention to your opponents' positions when you see them in order to duel with them and gain information about their strategy.
To find a specific player, you can first move towards the last known location of the player and then move randomly around the map.
Hanging around the center of the map and waiting for a player to come to you is not a good strategy for this environment.
After you gather information about your opponents' strategies, seek out opponents whose strategy you know and can exploit and play a counter-strategy.

---

### G.1.1 HYPOTHETICAL MINDS

---

**Subgoal Module Message**

---

**Current State Description:**
- Global Map Size: {map_size} grid (Walls are located at the boundaries of
the map and in other places that are invalid for move_to).
- Valid Locations for move_to: {movable_locations}
- Player Position: {player_position}
- Player Orientation: {player_orientation}
- Player Inventory (yellow, purple, blue): {player_inventory}
- Egocentric Observations Size: 11x11 grid around your agent. You
currently can observe the following based on your position and orientation:
- Observable Yellow Box Locations (format: ((x,y), distance from current
location)): {yellow_locations_with_distance}
- Observable Blue Box Locations: {blue_locations_with_distance}
- Observable Purple Box Locations: {purple_locations_with_distance}
- Observable Opponent Locations: {opponent_locations}
- Previously seen states from memory (format: ((x,y), step last observed,
distance from current location)): {self.memory_states}
**Execution Outcomes:**
{execution_outcomes}
**Error for extracting and executing actions from the response:**
{get_action_from_response_errors}
**Rewards:**
{rewards_str}
**Strategy Request:**
You are at step {step} of the game.
You have decided to execute a high-level strategy/target inventory in a
previous response given what you predicted your opponent will do.
Select subgoals in order to achieve the strategy, including first achieving
a target my_next_inventory: {self.hls_next_inventories}.
Once you achieve the target inventory, STOP picking up resources and
immediately seek out a duel with an opponent close to you that you can
exploit based on your hypothesis about their strategy and your current
inventory.
So once you've picked up about 5-7 resources in total, seek out a duel
to receive rewards, get more information about strategies, and reset your
inventory.
Here are your hypotheses about each player's strategy:
{self.opponent_hypotheses}
If you've generated a hypothesis about a player's strategy, you can use this
to inform your strategy about whether to interact with them or not.
Each strategy is paired with a value on how well it explains the data
observed so far, starting at 0.
A hypothesis is validated when its value is greater than:
{self.good_hypothesis_thr}.
Your task is to devise efficient action plans for player {self.agent_id},
reason through what the next subgoals should be given the state information.
Your response should be broken up into two parts:
1. Subgoal Plan – based on the current state and the high-level strategy
you previously specified above, decompose this strategy into a sequence of
subgoals and actions to efficiently implement this strategy. Think step by
step about this. This could be fairly long.
2. Action Plan – output this sequence of actions in the following Python
dictionary format, parsable by ast.literal_eval() starting with:

{{ 'action_plan': ['move_to((11, 7), (9, 5))', 'move_to((9, 5), (13, 5))'] }}

Example response 1, 2, and 3 are formatted similarly, detailing other
strategies and actions.

**ToM Module User Message 1**

An interaction with another player has occurred at step {step}, {self.interaction_history[self.last_played_id][-1]}.
**What was my opponent's likely inventory in the last round given the inventory I played and the reward received?**
Think step by step about this.  First think about what resource you had the most of in your inventory, and then think about which resource would beat that if you received a negative reward of -1 or worse or which resource would lose to yours if you received a positive reward of 1 or more.
If you received a small magnitude reward near 0 and in between (-1, 1), then your opponent may have played a similar inventory to you.
Then depending on the magnitude of the reward and the number of resources you played, you can infer the opponent's inventory and whether they played that strategy strongly (5+ of that resource) or weakly ( 3 of that resource).
An inventory of {'rock/yellow':  1, 'paper/purple':  1, 'scissors/blue':  1} is not possible because you need at least 2 resources of a type to play a duel.
Here are some example interactions to help you reason about how the reward function works:
'your_inventory':  {'rock/yellow':  3, 'paper/purple': 1, 'scissors/blue':  1}, 'rewards':  -2.285, 'possible_opponent_inventory':  {'rock/yellow':  1, 'paper/purple': 5, 'scissors/blue':  1}
'your_inventory':  {'rock/yellow':  5, 'paper/purple':  1, 'scissors/blue':  1}, 'rewards':  3.571, 'possible_opponent_inventory': {'rock/yellow':  1, 'paper/purple':  1, 'scissors/blue':  6}
'your_inventory':  {'rock/yellow':  1, 'paper/purple':  4, 'scissors/blue':  1}, 'rewards':  2.0, 'possible_opponent_inventory': {'rock/yellow':  3, 'paper/purple':  1, 'scissors/blue':  1}

In the 2nd part of your response, output the predicted opponent's inventory in following Python dictionary format, parsable by ast.literal_eval() starting with:  'possible_opponent_inventory': {'rock/yellow':  1, 'paper/purple':  1, 'scissors/blue':  5}
Example output:
Given that I last played a strong paper strategy with an inventory of {'rock/yellow':  1, 'paper/purple':  5, 'scissors/blue':  1} and received a reward of -3.428, I believe my opponent played a strong scissors strategy.
The reward suggests that my paper was beaten by their scissors, which means their inventory likely had a higher count of blue/scissors resources.
A possible inventory for them could be {'rock/yellow':  1, 'paper/purple':  1, 'scissors/blue':  5} or a similar distribution favoring scissors.

---

**ToM Module User Message 2**

```
Total Rewards: {rewards_str}
```
**Strategy Request:**
```
An interaction with another player has occurred at step {step},
{self.interaction_history[self.last_played_id][-1]}.
The total interaction history with this opponent is:
{self.interaction_history[self.last_played_id]}.
```
**If self-improvement is a focus:**
```
Here are your previous hypotheses about the algorithm this opponent
is playing: {self.top_hypotheses[self.last_played_id]}.
What is your opponent's likely policy given the inventories and
the reward function? Think step by step about this given the
interaction history.
If your previous hypotheses are useful, you can iterate and refine
them to get a better explanation of the data observed so far.
If a hypothesis already explains the data very well, then repeat
the hypothesis in this response.
They may be playing the same pure policy every time, a complex
strategy to counter you, or anything in between.
They are not necessarily a smart agent that adapts to your
strategy, you are just playing an algorithm.
Are you getting high positive or negative reward when playing the
same type of inventory? For example, getting high positive reward
every time you play many paper resources. If so, this opponent may
be playing a pure strategy and you can exploit this by playing the
counter strategy.
Once you have output a hypothesis about this opponent's strategy
with step by step reasoning, you can use the hypothesis to inform
your strategy.
In the 2nd part of your response, summarize your hypothesis
in a concise message following Python dictionary format,
parsable by ast.literal_eval() starting with: 'rock/yellow': 1,
'paper/purple': 1, 'scissors/blue': 5
{'Opponent_strategy': 'I think my opponent is always playing a pure
scissors strategy and collecting around 5 blue resources.'}
```
**Otherwise:**
```
What is this opponent's likely policy given the inventories and the
reward
function? Think step by step about this given the interaction
history.
They may be playing the same pure policy every time, a complex
strategy to
counter you, or anything in between.
They are not necessarily a smart agent that adapts to your
strategy.
Are you getting high positive or negative reward when playing the
same type of inventory? For example, getting high positive reward
every time you play many
paper resources. If so, this opponent may be playing a pure
strategy and you
can exploit this by playing the counter strategy.
Once you have output a hypothesis about this opponent's strategy
with step by step reasoning, you can use hypothesis to inform your
strategy.
In the 2nd part of your response, summarize your hypothesis in
a concise message following Python dictionary format, parsable
by ast.literal_eval() starting with: {'Opponent_strategy': 'I
think my opponent is always playing a pure scissors strategy and
collecting around 5 blue resources.'}
You will be prompted again shortly to select subgoals and action
plans to execute this strategy that achieves the target inventory,
so do not include that in your response yet right now.
```

**ToM Module User Message 3**

```
An interaction with self.last_played_id has occurred at step
{step}, The total interaction history with {self.last_played_id{
is:  You previously made the following guess about this player's
strategy:  Think step by step and predict what this opponent
will play the next time you interact with them.  Given the
above mentioned guess about the opponent's policy/strategy,
and the last inventory you played (if their strategy is
adaptive, it may not be), what is their likely inventory in
the next round.  In the 2nd part of your response, output
the predicted opponent's next inventory in following Python
dictionary format, parsable by `ast.literal_eval()` starting
with ```python.  Example response 1:  'Opponent_next_inventory':
Given that my opponent is playing a rock policy, I believe
their next inventory will be inventory={'rock/yellow':  5,
'paper/purple':  1, 'scissors/blue':  1}.  ```python {
'predicted_opponent_next_inventory':  {'rock/yellow':  5,
'paper/purple':  1, 'scissors/blue':  1} } ``` Example response 2:
'Opponent_next_inventory':  Since my guess is that this player is
playing a scissors policy, I predict that their next inventory will
be {'rock/yellow':  1, 'paper/purple':  1, 'scissors/blue':  5}.
```python { 'predicted_opponent_next_inventory':  {'rock/yellow':  1,
'paper/purple':  1, 'scissors/blue':  5} } ``` Example response
3:  'Opponent_next_inventory':  Since my opponent is following
a paper strategy, I predict their upcoming inventory will be
inventory={'rock/yellow':  1, 'paper/purple':  5, 'scissors/blue':
1}.  ```python { 'predicted_opponent_next_inventory':
{'rock/yellow':  1, 'paper/purple':  5, 'scissors/blue':  1} }
```

**ToM Module User Message 4**

An interaction with self.last_played_id has occurred at step {step},
The total interaction history with self.last_played_id is:   The
total interaction history overall is:  .  You previously made
the following guesses about all the other players' strategies:
possible_opponent_strategy.  High-level strategy Request:  Provide
the next high-level strategy for your player self.agent_id.  This
response should include step by step reasoning in parts 1-3 about
which strategy to select based on the entire interaction history
in the following format:  1.  'Opponents_to_seekout':  Given the
hypotheses about your opponent's strategies and their values,
which players should you seek out to duel with next and why?  If
possible, select opponents you have a good hypothesis about so you
can exploit it and maximize your reward.  Try to select multiple
players if possible as one player might be hard to find or is
respawning.  Are you noticing any patterns across the population
as a whole?  2.  'Opponent_next_inventory':  Given the above
mentioned guess about the opponent's policy/strategy what is their
likely inventory in the next round.  3.  'My_next_inventory':
Given the opponent's likely inventory in the next round, what
should your next inventory be to counter this?  4.  In the
4th part of your response, output the opponent to seekout,
the predicted opponent's next inventory, and your next
inventory in following Python dictionary format, parsable by
`ast.literal_eval()` starting with ```python.  Example response
1:  1.  'Opponent_to_seekout':  Given that I am fairly certain
that player_1 and player_5 is playing a rock policy, I believe I
should seek out either player_1 or player_5 to duel with next.  2.
'Opponent_next_inventory':  Given that these opponents are playing a
rock policy, I believe their next inventory will be something like
inventory={'rock/yellow':  5, 'paper/purple':  1, 'scissors/blue':
1}.  3.  'My_next_inventory':  Given that these players are playing
a rock policy, I believe my next inventory should be a paper policy
inventory={'rock/yellow':  1, 'paper/purple':  5, 'scissors/blue':
1}.  ```python { 'opponents_to_seekout':  ['player_1', 'player_5'],
'predicted_opponent_next_inventory':  {'rock/yellow':  5,
'paper/purple':  1, 'scissors/blue':  1} 'my_next_inventory':
{'rock/yellow':  1, 'paper/purple':  5, 'scissors/blue':  1} }
Example response 2:  1.  'Opponents_to_seekout':  Considering
all the interactions, player_2, player_3, and player_5 seem to
heavily favor the scissors/blue strategy with consistent picks.
Engaging either of these players could offer a high reward
opportunity.  2.  'Opponent_next_inventory':  Based on the observed
behavior of player_2, player_3, and player_5, it is likely they
will continue with a strong scissors/blue strategy, potentially
having an inventory of {'rock/yellow':  1, 'paper/purple':  1,
'scissors/blue':  5}.  3.  'My_next_inventory':  To effectively
counter the scissors/blue strategy, my optimal approach would be
to adopt a rock/yellow strategy.  An inventory geared towards this
would be {'rock/yellow':  5, 'paper/purple':  1, 'scissors/blue':
1}.  ```python { 'opponents_to_seekout':  ['player_2', 'player_3',
'player_5'], 'predicted_opponent_next_inventory':  {'rock/yellow':
1, 'paper/purple':  1, 'scissors/blue':  5}, 'my_next_inventory':
{'rock/yellow':  5, 'paper/purple':  1, 'scissors/blue':  1} }

## G.2 RUNNING WITH SCISSORS: REPEATED

The prompts for Running with Scissors Repeated remain very similar as Running with Scissors: Arena with the modification of the prompts being in a 2 player setting.

## G.3 PRISONER'S DILEMMA: REPEATED

**System Message**

You are Agent {self.agent_id} in the two player 'prisoner_dilemma_in_the_matrix' Melting Pot multiagent reinforcement learning environment that is an 23x15 (x by y) grid with resources to collect and walls to navigate around. Your goal is to maximize reward. Players can move around the map and collect resources of 2 discrete types corresponding to cooperate and defect strategies – Green box = cooperate – Red box = defect. You and another agent will collect resources and compare inventories in an encounter in repeated rounds. In addition to movement, the agents have an action to fire an "interaction" beam which initiates a duel to compare inventories and distribute rewards. All players carry an inventory with the count of resources picked up since last respawn and for each respawn start with an inventory of 1 resource each. This inventory is visible in the state with the key 'inventory'. Here is how the payoff matrix in the generic prisoner dilemma works:

- If both players cooperate, they both receive a reward of around 3,
- If one player cooperates and the other defects, the defector receives a reward of around 5 and the cooperator receives a reward of around 0,
- If both players defect, they both receive a reward of around 1.

The nuance of 'prisoner_dilemma_in_the_matrix_repeated' is that the rewards are using the payoff rules of the classic prisoner dilemma, but the interaction is repeated and the rewards are distributed based on the inventory of resources collected by each player.

- If both agents cooperate, the one with more cooperate resources will receive a reward lower than the one with less cooperate resources,
- If one agent cooperates and the other defects, the more defect resources the defector has, the higher the reward for the defector,
- If both agents defect, the one with more defect resources will receive a higher reward than the one with less defect resources.

Your goal before each interaction is to try and infer what the other player will play and how their strategy over time is affected by your plays. You will only want to pick up one type of resource before an interaction. For example, the inventories {'cooperate/green': 1, 'defect/red': 1} and {'cooperate/green': 3, 'defect/red': 3} will both result in the same reward, so don't waste time collecting more than you need. Player 1 will also always only pick up one type of resource before an interaction. To play a strategy strongly, pick up at least 6 resources or more of only one color and then fire the interaction beam at the other player. To commit less strongly to a strategy, pick up around 2 resources of only one color and then fire the interaction beam at the other player. State Description: This environment is partially-observable, you can observe a 5x5 grid around your agent depending on your position and orientation (you can see more in front of you than behind). Previously seen states will be represented in memory, but note that these states could potentially be outdated. For example, the other agent could collect a resource that you previously saw. Given the partially-observable nature of the environment, you will need to explore the environment appropriately and select goals based on the information you've gathered. Also pay attention to Player 1's position when you see it in order to duel with them and gain information about their strategy. Your goal is to maximize reward attained over an entire episode, so keep in mind the long-term consequences of your actions. Look at events in a gestalt manner.

## G.4 COLLABORATIVE COOKING PROMPTS

---

**System Message for Collaborative Cooking Asymmetric**

You are Player {self.agent_id} in the Collaborative Cooking Asymmetric environment, the goal is to cook and deliver tomato soup dishes with a partner. The environment consists of a kitchen with a tomato dispenser, pots, delivery locations, and dish dispensers. Each agent (of 2) has access to specific parts of the kitchen and can perform actions like picking up ingredients, putting soup in a dish, and delivering cooked soup dishes. There is an impassable barrier in the middle of the kitchen that separates the agents' sides at x=4, where the pots are located. The goal is to work together with the other agent to efficiently cook and serve as many dishes of tomato soup as possible to maximize the collective reward. However, communication is not possible, so you must infer your partner's strategy from their actions and adapt accordingly to coordinate tasks. To cook tomato soup, 1. put 3 tomatoes in a pot, 2. pick up a dish when it is finished cooking, 3. put the cooked soup in a dish, and 4. deliver it to the delivery location. Your team receives a reward of 20 for each successfully delivered dish. Only interact with objects on your side of the kitchen. You can only hold one tomato at once. You cannot pick up a tomato from the tomato dispenser with another item like a dish in your hand. You need to pick up a dish before you pick up cooked soup from a pot. The environment is partially observable, and you can only see a 5x5 grid around your agent. You will be prompted at different points to provide high-level strategies and lower-level action plans to achieve them.
**Use these three functions for lower-level action plans:**

- move_to(src_coord, target_coord): Efficiently move agent from source coordinate to target coordinate. Only move to valid move_to locations where counters or objects are not present. Use sparingly.

- interact(target_coord): Move to and interact with the entity at the target coordinate, such as picking up ingredients or delivering dishes of cooked soup. To place an object down on a counter to free your hands, use interact(counter_coord). Mostly use this function.

- wait(target_coord): Wait for the pot at target_coord to finish cooking. Check the progress of the pots and only use valid locations where pots are present. You probably only want to use this when both pots are full to maximize efficiency.

Most of the time you will just want to use the interact function because it both moves to and interacts with objects, therefore all the cooking steps can be completed with the interact function. To put down an item to pick something else up, interact with a counter to free your hands. Do not put down items on the floor or the delivery location.

---

### G.4.1 HYPOTHETICAL MINDS

---

**High Level Strategy Message (in ToM Module of HM) for Collaborative Cooking Asymmetric**

**Strategy Request:**
You are at step {step} of the game.
Provide a strategy for agent {self.agent_id}.
Your response should outline a high-level strategy - what strategy do you want to take next and why?
Teammate's observed strategy: {self.teammate_strategy}
Think step by step about how to adapt to their behavior and maximize all resources and efficiency accordingly.
**This response will be shown to you in the future in order for you to select lower-level actions to implement this strategy.**
Example response:
High-level strategy: I want to focus on cooking tomato soup dishes.
You will be prompted again shortly to select subgoals and action plans to execute this strategy, so do not include that in your response yet.

---

---

**Subgoal Module Message**

**Strategy Request:**
You are at step {step} of the game.
Your task is to devise efficient action plans for agent
{self.agent_id}, reason through what the next subgoals should be
given the state information.
Your previously specified high-level strategy is:
{self.my_strategy}
Your response should be broken up into two parts:

1. **Subgoal Plan** – Based on the current state and the high-level
   strategy you previously specified, decompose this strategy
   into a sequence of subgoals and actions to efficiently
   implement this strategy. For every subgoal, think step by
   step about the best action function and parameter to use for
   that function. This could be fairly long.

2. **Action Plan** – Output this sequence of actions in
   the following Python dictionary format, parsable by
   ast.literal_eval() starting with:

   {{ 'action_plan': ['interact((5, 1))'] }}

Example response 1:
Subgoal Plan: Given the current state and my high-level strategy
to focus on cooking tomato soup dishes, I should:
Move to the tomato dispenser and pick up a tomato.

{{ 'action_plan': ['interact((5, 1))'] }}

Example response 2:
Subgoal Plan: Given the current state and my high-level strategy
to focus on delivering tomato soup dishes, I should:
Move to the dish dispenser and pick up a dish, then plate the
cooked soup.

{{ 'action_plan': ['interact((3, 4))', 'interact((4, 2))'] }}

Example response 3:
Subgoal Plan: Next I should move to the delivery location and
deliver the cooked soup.

{{ 'action_plan': ['interact((3, 1))'] }}

**Evaluate Action Outcomes/Self-Reflection**

**User Message Preamble:**
**If subgoal failed:**
You are an action plan evaluator.
The last subgoal included an interact action that failed.
Your task is to look at the subgoal the agent took, the state of
the environment before and after the subgoal,
and evaluate why the subgoal was unsuccessful and provide feedback
about what the agent should do next time.
We will next plan an entire new action plan, so suggest specific
action plans and action functions to use next when applicable.

**If subgoal succeeded:**
You are an action plan evaluator.
Your task is to look at the action plan the agent took, the state
of the environment before the plan and the state of the environment
after the plan,
and evaluate whether the action plan was successful, and if not,
provide feedback about what failed and what the agent should do
next time.
Take into account that your teammate could have influenced the
outcome of the subgoal in some circumstances.
Suggest specific action plans and action functions to use next when
applicable.

**Infer Teammate Strategy Message**

**User Message:**
Based on the observed actions of your teammate (player_1), what do
you think their strategy is?
Are they specializing in any specific activity or subtask?

Teammate's observed actions:
{self.teammate_actions}

Here are your previous hypotheses about the strategy your partner
is playing: {self.top_hypotheses}.

Think step by step and provide an analysis of their strategy, any
specialization you infer from their behavior, and their competence.
Then analyze how you can adapt your strategy to maximize efficiency
and coordination with your teammate.
Remember communication is not allowed.

**Predict Teammate Behavior Message**

```
A dish has been delivered at step {step}.
You previously guessed that your teammate's (player_1) policy is:
{possible_teammate_strategy}
Based on the proposed hypothesis about your teammate (player_1),
what do you think they will do next?
Output a concise label about your teammate's next behavior in the
following Python dictionary format, parsable by ast.literal_eval()
starting with:

python
{{ 'predicted_next_behavior': 'placing tomatoes into pot (4,2)' }}
```

**Evaluate Predicted Behavior**

```
A dish has been delivered at step {step}.
You previously guessed that your teammate's (player_1) would perform
this behavior in this round:  {predicted_next_behavior}
Here is the observed behavior of your teammate (player_1) in this
round:  {latest_teammate_actions}
Did your prediction match the observed behavior?
Concisely output True or False in the below Python dictionary
format, parsable by ast.literal_eval() starting with:

{{ 'evaluate_predicted_behavior': True }}
```

