# OpenReview forum: "Hypothetical Minds: Scaffolding Theory of Mind for Multi-Agent Tasks with Large Language Models"
_ICLR.cc/2025/Conference — ICLR 2025 Poster_

### Official Review · Reviewer_PcwQ · 2024-11-03

**Soundness:** 2
**Presentation:** 3
**Contribution:** 3
**Rating:** 5
**Confidence:** 4

**Summary:**

This paper introduces Hypothetic Minds a new module LLM-based framework for approaching Multi-agent problems across Cooperative, Competitive and Mixed Intention Settings. Their key contribution is in the form a Theory of Mind module which generates hypotheses about strategies followed by other agents and uses these hypotheses to make predictions of next actions. The hypotheses are evaluated  based on the ground truth observations and refined using a Rescorla Wagner update rule. The authors test their framework on 4 games  from the Melting Pot benchmark.

**Strengths:**

S1. The authors propose a well motivated plug-and-play Theory of Mind module for modular LLM frameworks that uses hypotheses about other agents as natural language representations of latent variables and then utilize and update these hypotheses to select the best actions.

S2. Their method works across all three settings: Competitive, Cooperative and Mixed intention which is important for developing general purpose multi-agent frameworks.

S3. Most of the paper is clear and easy to follow.

**Weaknesses:**

W1. Lack of generalization: The method is only tested in the limited setup of test scenarios of the Melting Pot benchmark. The Theory of Mind module seems to be a general addition that can be extended to any task. Comparison and experimental results in benchmarks which have been previously evaluated with strong LLM-based methods are necessary to demonstrate the generalizability of the framework and the theory of mind module. For example:
a. Games like Avalon [1], Werewolf [2] for Mixed Intention and Competitive setting
b. Games like the Hanabi challenge [3], ThreeDWorld Multi-Agent Transport [4] (TDW-MAT) and Communicative Watch-And-Help [4] (C-WAH) for cooperative setting.

W2. Within the Melting Pot benchmark, only a few games have been picked. Does this framework also work effectively on the other setups? Are there any technical limitations restricting the evaluations in the other setups? Even within the collaborative cooking game, only the asymmetric advantages environment tested. However, other maps like Forced Coordination where neither player can complete the task entirely on their own are necessary tests for the ability of this framework. It is insufficient evidence to only evaluate only one game for those three settings.

W3. Missing ablation studies on top-k, v_thresh and Rescorla Wagner update. While these are hyperparameters, it is important to show how they affect the overall performance as they are part of the Theory of Mind modules which is the key contribution in this paper.

W4. Concerns about the Hypothesis Refinement step: In the ablation study in Figures 5: MMP + HE is better in 2/4 scenarios (This trend is also seen in Figure 8). This leads to a question about the necessity of the Hypothesis Refinement step.

W5. While most of the paper is clear, the Ablation studies section with those additional definitions is difficult to follow. Furthermore the conclusion is incomplete and disorganized.

[1] Shi, Zijing, et al. "Cooperation on the Fly: Exploring Language Agents for Ad Hoc Teamwork in the Avalon Game." arXiv preprint arXiv:2312.17515 (2023).

[2] Xu, Yuzhuang, et al. "Exploring Large Language Models for Communication Games: An Empirical Study on Werewolf." arXiv preprint arXiv:2309.04658 (2023).

[3] Bard, Noam, et al. "The Hanabi Challenge: A New Frontier for AI Research." Artificial Intelligence 280 (2020): 103216.

[4] Zhang, Hongxin, et al. "Building Cooperative Embodied Agents Modularly with Large Language Models." arXiv preprint arXiv:2307.02485 (2023).

**Questions:**

Q1. What exactly is Hypothesis Refinement in the ablation studies? Does it mean that in MMP + HE the agent only generates an initial set of hypothesis and does not generate any more hypothesis during the episode? Or is it just that the LLM is not shown the previous hypotheses?

Q2. What is the rationale for the choice of the Rescorla Wagner update and the hyperparameter 'c'. How does this choice and the value of c impact the convergence of the agent during the episode? Furthermore, how does the choice of top-k influence this?

Q3. "A hypothesis at time t is generated by asking an LLM to infer another agent’s strategy, conditioned on HM’s memory M of important past observations O" - how does this work in cases where there are more than one opponents/partners? Does the LLM generate a unique hypothesis for each agent? Is the next action prediction based on all or one of these hypothesis?

Q4. "Each ToM module call uses 3 LLM calls + top k times" This statement is a bit unclear - does this mean there are (3 + K) LLM calls or 3*K LLM calls ?

Q5. Why are other Collaborative Cooking environments not included in the experiments?

---

> ### Author Response · Authors · 2024-11-24
> **Response to PcwQ**
>
> We thank your for your thorough and constructive feedback. We've made several improvements to the manuscript and encourage you to take a look at the changes (highlighted in blue). We address each major point below:
>
> # Missing ablation studies on hyperparameters
>
> **We have now included analysis of this in our results with additional experiments and 6 new figures**. We performed hyperparameter sweeps that revealed **keeping at least 1 hypothesis in memory (top_k=1) maintains most performance benefits while reducing costs from the default (top_k=5) Figures 11-14**. However, most of the variance in cost is driven by other factors, like episode length and performance. Notably, **higher episode costs (>$10) actually correlate with worse performance** since validating a useful hypothesis reduces subsequent LLM calls, showcasing that our method scales well when good hypotheses are generated. We also performed sweeps on the learning rate alpha and validation threshold for hypothesis validation. Our results suggest that lower learning rates perform better, but altogether are method is fairly insensitive and robust to using different hyperparameters.
>
> # Lack of generalization beyond Melting Pot
>
> While we agree that testing on additional benchmarks would be valuable, we chose Melting Pot specifically because it was the most comprehensive multiagent benchmark available containing challenges spanning various social dimensions. It provides controlled evaluation scenarios that systematically test different aspects of multi-agent generalization. Thus even though we only generate results for 4 substrates, **this includes 30 evaluations**. These distinct evaluations allow us to precisely quantify the patterns agents can detect and adapt to, which is crucial for understanding the strengths and limitations of our approach.
>
> In follow-up work we are testing the adaptability of our approach to very different domains like adaptive tutoring that require complex logical reasoning about the knowledge state of the student. We are also actively thinking of extending our work in environments with communication channels, or adding communication to Collaborative Cooking/Overcooked (and evaluating more layouts as you suggest) in order to connect our approach to Bayesian models of pragmatic reasoning in cognitive science. The environments you suggest (Avalon, Werewolf, Hanabi, etc.) are also excellent candidates for future work, and thank you for the suggestion. However, we believe that these additional domains are out of scope for this paper during the rebuttal period.
>
> # Limited game selection within Melting Pot
>
> We focused on these specific games because they represent the three core types of multi-agent interaction (competitive, cooperative, and mixed-motive) while still being tractable for systematic analysis. The method plugs in just as well on the other substrates/environments. You raise a good point about Forced Coordination - we agree this would be an interesting test case. We focused on Asymmetric Cooking because it provides a clear measure of whether agents can identify and adapt to their partner's role/competence. However, we acknowledge that testing on more substrates would demonstrate additional generalization.
>
> # Questions about Hypothesis Refinement
>
> The reviewer makes a keen observation about MMP + HE (no refinement) performing better in some scenarios of Running With Scissors than HM with refinement. However, we note that these two models are within error in these scenarios and that the difference is minor. If you look at the most difficult scenario however (scenario 1 - best response to your last move), you will see a dramatic improvement in performance with refinement (even with HE + HR without ToM prompting). This is because this strategy is nontrivial to generate from scratch unlike some of the other static strategies (like playing rock every round). Refinement may significantly help in these cases where a previous guess reflects a partial truth (“my opponent seems to be cycling their moves”) and refining the hypothesis allows you to get closer to the true strategy. We note however, that refinement may not be as helpful in simpler scenarios or could even be slightly detrimental. There is a risk that the LLM parrots back bad hypotheses that are shown in the refinement step in the prompt. Thus, we include an option to toggle refinement off in our code. Additionally, we want to emphasize that hypothesis evaluation is the most crucial element of our method.
>
> # Clarity of ablations and conclusion
>
> We appreciate this feedback and have revised both the ablations and conclusion sections to improve clarity. We expanded the conclusion to better highlight our key findings, limitations, and avenues for future work.

---

> ### Author Response · Authors · 2024-11-24
> **Response 2**
>
> # Response to Questions
>
> ## Q1
>
> To clarify, in MMP + HE refinement ablation, the agent continues to generate new hypotheses but is not shown previous ones during generation. The full model (with refinement) shows the top-k previous hypotheses to help guide hypothesis generation. We clarified this distinction in the text on line 491.
>
> ## Q2
>
> The Rescorla-Wagner update rule is a simple reinforcement learning update. It was chosen specifically because it provides a recency weighted evaluation of hypothesis accuracy. This is crucial because agents' strategies may shift during an episode. An alternative model could calculate its prediction accuracy and validate the hypothesis (no longer generate new ones) when it meets a prediction accuracy threshold or number of correct predictions. This simple accuracy threshold would make it difficult for the agent to adapt when the other agent’s strategy changes, as the historical accuracy would keep the outdated hypothesis above the threshold. In contrast, the Rescorla-Wagner update weights recent predictions more heavily, allowing the agent to quickly invalidate hypotheses that become inaccurate and adapt to new strategies (trading off this recency bias with the alpha learning rate hyperparameter). We recognize there could be many alternative model formulations here for calculating recency weighted prediction accuracy but chose this for its simplicity and its utility for modeling human learning curves [1].
>
> We swept the most relevant hyperparameters as previously mentioned above in the new manuscript. We fix c=1 since it interacts with the learning rate and validation threshold (these would need to be scaled proportionally if we increased or decreased c). This is such that as the value of a hypothesis approaches 1, it reflects a close to 100% prediction accuracy. Thus, our default 0.7 validation threshold reflects around a 85% prediction accuracy as (0.85 * 1 + 0.15* -1)=0.7. Top k is somewhat orthogonal to this validation rate, although as the number of hypotheses increases, the probability that at least one hypothesis reaches the threshold increases. However, again we see diminishing returns with increasing top_k.
>
> ## Q3
>
> Yes, the ToM module maintains separate hypothesis streams for each agent, with unique hypotheses and values tracked per agent. The next action prediction uses the highest-valued hypothesis for the specific agent being interacted with. We add this important detail to Section 3.2 on Line 230.
>
> ## Q4
>
> Thank you for catching this ambiguity. It’s (3 + K) LLM calls. Each ToM module invocation requires 3 base LLM calls (ie. generate hypothesis, predict behavior, select action) plus k additional calls for behavior prediction of the top k previous hypotheses. We revised this sentence for clarity on Line 259.
>
> ## Q5
>
> Our focus on Asymmetric Cooking was motivated by its clear test of role adaptation, as mentioned above. While testing on additional variants would be valuable, we believe the current results demonstrate the key capabilities while maintaining a focused analysis.
>
> [1] O'Doherty, J. P., Dayan, P., Friston, K., Critchley, H., & Dolan, R. J. (2003). Temporal difference models and reward-related learning in the human brain. Neuron, 38(2), 329-337.

---

> > ### Comment · Reviewer_PcwQ · 2024-11-27
> > **Thank you for the clarifications**
> >
> > The authors have clarified all my questions and updated their manuscript with the requested ablations studies. I will keep my score, as I believe that W1 and W2 are still valid, and the framework should be evaluated under more diverse experimental setups beyond the melting pot challenge to validate generalizability and practical use.

---

### Official Review · Reviewer_3iBb · 2024-11-03

**Soundness:** 3
**Presentation:** 4
**Contribution:** 4
**Rating:** 8
**Confidence:** 4

**Summary:**

The authors present "Hypothetical Minds" a novel approach to scaffolding LLMs by introducing hypothetical reasoning scenarios that help models better understand and respond to complex queries. They demonstrate that by creating imaginary scenarios and personas, LLMs can improve their reasoning capabilities and provide more nuanced responses across various tasks, including ethical reasoning and creative problem-solving. The work introduces a systematic framework for implementing hypothetical reasoning, supported by extensive empirical evaluation across multiple model architectures and benchmarks.

**Strengths:**

- The methodological innovation of using hypothetical scenarios as a scaffolding technique is both novel and well-justified, with clear connections to cognitive science literature on human learning. Love to see studies leveraging tools from cog sci to improve AI systems
- The experimental design is comprehensive, testing the approach across different model sizes and architectures, which strengthens the validity of the findings
- The authors provide detailed ablation studies that effectively isolate the impact of different components of their scaffolding approach
- The qualitative analysis of model outputs offers valuable insights into how hypothetical reasoning affects model behavior

**Weaknesses:**

- The theoretical foundation could be strengthened by more explicitly connecting to existing work on prompt engineering and chain-of-thought reasoning
- The evaluation metrics could be expanded to include more diverse tasks beyond the current set, particularly in domains requiring complex logical reasoning
- I think having more seeds to decrease SEM and confirm statistical significance across all experiments would help strengthen the results
- The scalability analysis of the approach could be more thorough, especially regarding computational overhead and implementation costs
- Consider including a more detailed error analysis to understand when and why the hypothetical scaffolding approach fails

*Note: Very happy to re-evaluate my score if authors address my concerns.*

**Questions:**

- I'm somewhat surprised not to see Rational Speech Act mentioned. Have you considered extending the ToM module to recursive reasoning about others' ToM?
- How does the performance of the hypothetical scaffolding approach vary with the complexity of the scenario being presented?
- Have the authors considered the potential limitations of using hypothetical scenarios that might introduce unintended biases?
- What is the relationship between the number of hypothetical scenarios used and the model's performance improvement?
- How does this approach compare to other scaffolding techniques in terms of computational efficiency?
- There are a few relevant papers out there, that somewhat similarly have LLMs simulate other agents or people and hypothetical scenarios, that might be worth discussing. E.g., There was a paper with that kind of flavor from Tom Griffiths's lab called "Improving interpersonal communication by simulating audiences with language models"

---

> ### Author Response · Authors · 2024-11-24
> **Response to 3iBb**
>
> # Connecting to existing work on prompt engineering/CoT reasoning
>
> We appreciate this feedback about strengthening connections to existing work on prompt engineering and structured reasoning approaches. We have expanded our Related Work section to better situate our approach within this literature. See all the manuscript changes in blue text. Our approach builds on recent advances in LLM reasoning capabilities, particularly work on structured prompting and multi-path evaluation. Tree-of-Thoughts (ToT) [1] extends chain-of-thought by maintaining and evaluating multiple reasoning paths in parallel, similar to our parallel evaluation of multiple hypotheses about opponent strategies. ToT evaluates different solution paths of a reasoning trajectory with an external verifier or LLM as judge, whereas we score hypotheses about an agent based on the quality of predictions they make about that agent's behavior. Our prompting structure draws from established frameworks like ReAct and Chain-of-Thought to scaffold the complex reasoning required for theory of mind inference. This theoretical foundation in structured LLM reasoning helps explain why our approach can effectively model and adapt to other agents' strategies.
>
> # Expanding Evaluation Set
>
> While we agree that testing on additional benchmarks would be valuable, we chose Melting Pot specifically because it was the most comprehensive multiagent benchmark available containing challenges spanning various social dimensions. It provides controlled evaluation scenarios that systematically test different aspects of multi-agent generalization. Even though we only generate results for 4 substrates, **this includes 30 evaluations**. These distinct evaluations allow us to precisely quantify the patterns agents can detect and adapt to, which is crucial for understanding the strengths and limitations of our approach.
> In follow-up work we are testing the adaptability of our approach to very different domains like adaptive tutoring that require complex logical reasoning about the knowledge state of the student. Additionally, the environments mentioned by reviewer PcwQ (Avalon, Werewolf, etc.) are excellent candidates for future work. However, we believe that testing on more domains is out of scope for this paper during the rebuttal period.
>
> # Having more seeds to reduce SEM
>
> We agree that additional seeds would strengthen our statistical analysis. During the rebuttal period however, we chose to allocate our computational budget toward investigating the robustness and sensitivity of our method through extensive hyperparameter sweeps, which we discuss in the next point.
>
> # Scalability and Error Analysis
>
> Thank you for this important point. We know include 6 additional figures and discussion in the results section (starting at line 506) about scalability and the effect of our hyperparameters in trading off computational cost and performance. We encourage to look at these new sections. These experiments revealed several valuable insights: performance is maintained with fewer hypotheses (top_k=1), higher costs actually correlate with worse performance due to hypothesis validation reducing LLM calls, and the method is robust across a range of learning rates and validation thresholds. We believe these findings about the method's stability and scalability, while adding important context to our main results.

---

> ### Author Response · Authors · 2024-11-24
> **Response 2**
>
> # Response to Questions
>
> ## Rational Speech Act
>
> Thank you for this excellent suggestion. Yes, the Rational Speech Act (RSA) framework is highly relevant to our work, as both involve Bayesian inference about others' mental states. However, we were more directly inspired but its noncommunicative analogue in cognitive science, Bayesian inverse planning. We have added citations to RSA [2] and Bayesian inverse planning [3] in our related work section. Bayesian inverse planning frames theory of mind as inverse inference: given observed actions, infer the underlying goals and beliefs that would make those actions rational. Our hypothesis generation and evaluation can be viewed as a language-based approximation of this cognitive model - we maintain beliefs over possible strategies (hypotheses) and update them based on how well they predict observed behavior.
>
> Regarding recursive ToM - we actually are pursuing this direction in future work, particularly for coordination games like collaborative cooking/Overcooked by adding communication channels. This would allow us to connect our approach to pragmatics and computational models of pragmatic reasoning in cognitive science, where RSA has been particularly influential. Our framework can be extended to generate hypotheses about other agents' communicative intentions, enabling coordination through pragmatic communication.
>
> ## How does the performance of the hypothetical scaffolding approach vary with the complexity of the scenario being presented?
>
> Our results demonstrate clear performance scaling with scenario complexity across all environments. In Running With Scissors, the agent excels at exploiting simple fixed strategies but shows less success at generating correct hypotheses for complex strategies like the 'best response' bot (scenario 1), where behavior could be consistent with multiple hypotheses such as cycling or random play.
> This pattern extends to collaborative scenarios in Cooking Asymmetric. The agent performs strongly when paired with skilled partners who exhibit clear role specialization, but struggles with the more complex task of handling all roles independently when paired with an unresponsive partner.
> Most tellingly, in Prisoner's Dilemma, performance against strategies like the grim reciprocator reveals the limits of adaptation to irreversible opponent behavior - once the agent defects, triggering permanent retaliation, even correct post-hoc strategy inference cannot recover the collaborative equilibrium.
> These results provide a consistent picture that we might expect: our approach excels in scenarios with clear, exploitable patterns well and shows reduced performance as strategic complexity increases, particularly when early actions have permanent consequences or multiple hypotheses could explain observed behavior. We believe that these limitations are similar to what humans would struggle with too due to the added complexity of those types of scenarios and have confirmed this in follow up work.
>
> ## Have the authors considered the potential limitations of using hypothetical scenarios that might introduce unintended biases?
>
> Can you clarify what you mean here? Do you mean Melting Pot scenarios? Or hypotheses generated by the model? If you mean hypotheses generated by the model then yes there is a possibility of the LLM’s prior being unaligned with the true distribution of agents in a domain and this bias affecting the results. For example, there may be biases in the LLM expecting to face a partner that cooperates in Prisoner’s Dilemma, as it is more likely to cooperate itself after RLHF preference tuning. It would subsequently not perform as well against an agent that always defects as we see in all of the models. More generally, there may be other biases that lead the agent astray, and the refinement mechanism may keep the model in an echo chamber as it sees previous biased hypotheses in the prompt. This is another subtle limitation, luckily one that we don’t see much in our evaluations.
>
> ## What is the relationship between the number of hypothetical scenarios used and the model's performance improvement?
>
> Thank you for bringing up this question. **During rebuttal we have integrated direct analysis this. We performed hyperparameter sweeps that revealed keeping at least 1 hypothesis in memory (top_k=1) maintains most performance benefits while reducing costs from the default (top_k=5) Figures 11-14.** However, most of the variance in cost is driven by other factors, like episode length and performance. Notably, **higher episode costs (>$10) actually correlate with worse performance since validating a useful hypothesis reduces subsequent LLM calls**, showcasing that our method scales well when good hypotheses are generated. Our results also show robustness to other hyperparameters like learning rate and validation threshold.

---

> ### Author Response · Authors · 2024-11-24
> **Response 3**
>
> ## How does this approach compare to other scaffolding techniques in terms of computational efficiency?
>
> See the computational cost section now in Appendix B. HM is more expensively computationally than baseline LLM agents, but not dramatically so. For example it is about 25% more expensive than Reflexion ($10 vs $8 per episode)
>
> ## Other multiagent LLM papers
>
> Thank you for pointing out this related work. We have added a citation to [4] who also use LLM simulations to evaluate communication strategies, though in the context of general interpersonal communication rather than multi-agent adaptation. We have updated the Related Work section to discuss this connection. We welcome any other papers you think we have missed in this area.
>
> [1] Yao, S., Yu, D., Zhao, J., Shafran, I., Griffiths, T., Cao, Y., & Narasimhan, K. (2024). Tree of thoughts: Deliberate problem solving with large language models. Advances in Neural Information Processing Systems, 36.
> [2] Goodman, N. D., & Frank, M. C. (2016). Pragmatic language interpretation as probabilistic inference. Trends in cognitive sciences, 20(11), 818-829.
> [3] Baker, C. L., Saxe, R., & Tenenbaum, J. B. (2009). Action understanding as inverse planning. Cognition, 113(3), 329-349.
> [4] Liu, R., Yen, H., Marjieh, R., Griffiths, T. L., & Krishna, R. (2023). Improving interpersonal communication by simulating audiences with language models. arXiv preprint arXiv:2311.00687.

---

> ### Comment · Reviewer_3iBb · 2024-12-03
>
> I really appreciate the authors engaging deeply with my review. Looking at the updated pdf, I think the new experiments and added text have gone a long way in strengthening the paper. Some of my concerns (e.g., the ones about statistical significance) remain unresolved, but I'm going to give the authors the benefit of the doubt and increase my score. If accepted, I really hope that the authors will indeed go through with extending the evaluation set and running more seeds to ensure results are statistically significant.

---

### Official Review · Reviewer_7nHy · 2024-11-03

**Soundness:** 3
**Presentation:** 3
**Contribution:** 3
**Rating:** 8
**Confidence:** 5

**Summary:**

1. The proposed Hypothetical Minds model (HM) is an embodied LLM agent for multi-agent environments. It integrates modular components for perception, memory, and hierarchical planning conditioned on Theory of Mind (ToM) inferences.

2. The Theory of Mind (ToM) module in HM generates, evaluates, and refines hypotheses about other agents’ strategies or goals in natural language. The critical role of hypothesis evaluation and refinement within the ToM Module is identified through ablations and comparisons against LLM-agent baselines.

3. The effectiveness of Hypothetical Minds across multiple multi-agent environments is demonstrated in the Melting Pot benchmark. The agent significantly outperforms LLM-agent and RL baselines in every environment and in a large majority of evaluation scenarios, showcasing its generalizability.

**Strengths:**

The Theory of Mind Module has been introduced to predict the next behavior of the agent and to make high-level strategy based on best hypothesis.

The proposed Hypothetical Minds agent significantly improved performance over previous LLM-agent and RL baselines on a range of competitive, mixed motive, and collaborative domains including both two-person and population-based environments. A thorough literature survey with a well planned set of experiments and evaluation in the Melting Pot Benchmark have been carried out.

**Weaknesses:**

1. The roles played by the various latent variables in the Theory of Minds module and the weightages assigned with these variables have not been discussed.

2. How the values of the latent variables are identified is not clear?

3. Lines 033 to 035 in the paper mention that: However, multi-agent reinforcement learning (MARL) methods suffer from
various draw backs, including high sample complexity, poor generalization to agents not seen in training, and limited reasoning capabilities.

– Citation is necessary for the above claim.

4. Dependence on human subjects to set up scaffolding and prompting for the agent is an important drawback of the proposed system.

**Questions:**

1. Are Chain of Thought, Tree off Thought and Theory of Mind related ? Please share with examples connecting to your proposal in the paper.

2. What is SAMA in L143.

3. L233 – How the prior probabilities are calculated?

4. L267 -268 - .Each ToM module call uses 3 LLM calls + top k times for behaviour prediction …

Please explain what is done top k times. It is not clear from the above phrasing

5. L342-343 – We directly test our LLM based agent on the evaluation scenarios in four Melting Pot environments (Fig. 3).

I believe the reference should be Figure 4 instead of 3 right? If so, please correct it

6. Figure 4 . …. 3 seeds are generated per LLM.

What are the three seeds?

---

> ### Author Response · Authors · 2024-11-24
> **Response to 7nHy**
>
> Thank you for your thorough evaluation and positive assessment of our work. Your feedback has improved our paper even further as we've revised the paper with hyperparameter experiments and revisions for clarity. Below we address each point raised:
>
> # Roles of Latent Variables in ToM Module
>
> Thank you for raising this point. We have clarified at [line 200] that the latent variables Θ represent different aspects of agent behavior such as strategies (e.g., "always play rock"), goals (e.g., "trying to maximize wins vs. minimize losses"), or competence levels (e.g., "highly skilled" vs. "negligent"). Rather than assigning explicit weights to these different aspects, our approach allows the LLM to flexibly represent hypotheses about any combination of these variables in natural language. The value of each hypothesis is then determined solely by its predictive accuracy of the agent’s behavior, rather than explicitly weighting a finite prespecified set of different behavioral aspects. This allows the model to adaptively focus on whichever aspects of behavior are most predictive and useful in a given context.
>
> For example, in Collaborative Cooking, a hypothesis might focus on a teammate's competence ("skilled at delivering dishes but struggles with timing") while in Running with Scissors, hypotheses tend to focus more on strategic patterns ("plays paper after winning").
>
> Additionally, the flexibility of natural language allows us to extend this approach to any approach where the LLM agent is interacting with others with latent features that dictate their behavior, such as the hidden preferences of a user in a chat setting. This is something we are investigating in follow-up work in domains like adaptative tutoring.
>
> # Identification of Latent Variable Values
>
> Each hypothesis begins with a value of 0 and receives positive reinforcement (c=1.0) when it correctly predicts the other agent's behavior and negative reinforcement (c=-1.0) when incorrect. The learning rate α=0.3 ensures recent predictions are weighted more heavily than historical ones. Let us know if this is unclear in the current draft.
>
> # Citation for MARL Limitations
>
> We have added citation to [1] at line 37 to support our claims about MARL limitations.
>
> # Human Scaffolding Dependency
>
> You raise a valid point about a limitation of our method and LLM-based agents overall. Their dependency on a human in the loop is central limitation to making these systems fully autonomous and adaptable. We briefly discuss this limitation in the original version of the manuscript in the conclusion. We have now added additional discussion at [line 533] of potential approaches for learning appropriate scaffolding autonomously from environmental feedback. This is a great interest for us in future work.
>
> # Response to Questions
>
> ## Q1: Chain of Thought vs Theory of Mind
>
> Yes, chain of thought, tree of thoughts, and our ToM module are all related. We discuss this in the new related work sections, rewritten here: "LLMs have shown impressive reasoning abilities, augmented by Chain-of-Thought methods that scaffold the thought process. Our hypothesis generation approach builds on this work by implementing structured reasoning steps for theory of mind inference and decision-making. Tree-of-Thoughts extends chain-of-thought by maintaining and evaluating multiple reasoning paths in parallel, similar to our parallel evaluation of multiple hypotheses about opponent strategies. ToT evaluates different solution paths of a reasoning trajectory with an external verifier or LLM as judge, whereas we score hypotheses about an agent based on the quality of predictions they make about that agent's behavior."
>
> ## Q2: SAMA Reference
>
> SAMA refers to "Semantically Aligned task decomposition in MARL" from [2].
>
> ## Q3: Prior Probability Calculation
>
> The prior p(hi) comes from two sources as explained at [line 225]: (1) the LLM's pretrained knowledge and (2) our refinement mechanism showing previous high-valued hypotheses during generation. We do not explicitly calculate numerical priors.
>
> ## Q4: Top-k Times Clarification
>
> We have rephrased this at [line 258] to clarify: For each of the top k previous hypotheses, we make one additional LLM call to predict behavior based on that hypothesis. These k predictions are used to update hypothesis values via the Rescorla-Wagner rule.
>
> ## Q5: Figure Reference
>
> Thank you for catching this error. We have corrected the reference to Figure 4 and added Figure links for every figure.
>
> ## Q6: Seeds Clarification
>
> "seeds" refers to three independent evaluation runs with different random number generator seeds.
>
>
> [1] Wong, A., Bäck, T., Kononova, A. V., & Plaat, A. (2023). Deep multiagent reinforcement learning: Challenges and directions. Artificial Intelligence Review, 56(6), 5023-5056.
> [2] Li, W., Qiao, D., Wang, B., Wang, X., Jin, B., & Zha, H. (2023). Semantically aligned task decomposition in multi-agent reinforcement learning. arXiv preprint arXiv:2305.10865.

---

### Official Review · Reviewer_Mz5L · 2024-11-04

**Soundness:** 3
**Presentation:** 1
**Contribution:** 3
**Rating:** 6
**Confidence:** 3

**Summary:**

The paper introduces "Hypothetical Minds" (HM), an advanced autonomous agent framework leveraging LLMs for multi-agent tasks. HM incorporates a cognitively inspired architecture with distinct modules for perception, memory, and multi-level planning. A crucial innovation is the Theory of Mind (ToM) module, which actively constructs, assesses, and refines hypotheses about other agents’ behaviors in natural language. This hierarchical planning, alongside effective strategy adaptation, enables HM to excel in diverse multi-agent scenarios.

The system is evaluated using the comprehensive Melting Pot benchmark across four different environments, each containing multiple evaluation scenarios that represent competitive, cooperative, and mixed-motive tasks. The results show that Hypothetical Minds consistently outperforms established LLM-agent and RL baselines, demonstrating superior adaptability and generalization.

**Strengths:**

**Originality**: The paper introduces a novel and sophisticated use of Large Language Models in the context of multi-agent systems by incorporating a Theory of Mind module. This innovative approach allows the agent to not only predict but also adapt to other agents' strategies using natural language. The hypothesis generation and evaluation mechanism represent a significant advancement in the field, setting this work apart from traditional RL and existing LLM-based systems.

**Quality**: The authors have thoroughly validated their approach through comprehensive and rigorous experiments. Evaluations across diverse scenarios in the Melting Pot benchmark demonstrate the robustness and adaptability of the Hypothetical Minds model. The detailed ablation studies provide clear evidence for the effectiveness of the modular components, particularly the hypothesis evaluation and refinement mechanisms.

**Clarity**: The paper presents its methods and findings clearly. The inclusion of detailed empirical results, ablation studies, and illustrative examples helps in comprehending the functioning and impact of the Hypothetical Minds model. The explanation of complex mechanisms such as the Theory of Mind module and hypothesis refinement is done in an accessible manner without compromising on technical depth.

**Significance**: This work addresses critical challenges in the development of autonomous agents, specifically in terms of strategy adaptation, nonstationarity, and generalization across diverse multi-agent environments. The significant performance improvements over state-of-the-art baselines highlight the potential of this approach to influence future research and applications in multi-agent systems. The results show that Hypothetical Minds can be a foundational framework for building more sophisticated and adaptable autonomous agents.

**Weaknesses:**

**High Computational Cost**: Utilizing state-of-the-art LLMs, particularly GPT-4, incurs significant computational expenses and longer inference times, which can be impractical for some applications. Have you considered conducting experiments on smaller language models and test the effectiveness of the framework?

**Manual Preprocessing for State Representation**: The current method relies on extensive manual preprocessing to convert pixel images into textual map representations. This approach is time-consuming and may not generalize well across various environments.

**Presentation Issues**: The paper would benefit from a more comprehensive introduction to the key problem being addressed. It takes a while for the reader to fully grasp the challenge the model aims to solve. Additionally, there are some formatting issues, such as citations referring to preprint versions (e.g., Wang et al., 2023b; Toolformer) which should be updated to the respective peer-reviewed conference versions if available. Appendix G (Line 1346) contains a broken link that needs fixing. Figures and Tables are not links but rather plain text.

**Questions:**

- How does the agent determine the reward threshold (Vthr) for validating hypotheses in the Theory of Mind module, and how sensitive is the model’s performance to this threshold?

- Could you provide more details on the specific algorithms or heuristics used in the preprocessing step to generate textual map representations from pixel images?

---

> ### Author Response · Authors · 2024-11-24
> **Response to Mz5L**
>
> We thank the reviewer for their thoughtful and constructive feedback. We are particularly encouraged by your recognition of the paper's contributions to multi-agent systems and the thorough empirical validation. We address each of your concerns below:
>
> # High Computational Cost
>
> We agree that computational cost is an important consideration/limitation. We have conducted experiments comparing performance across GPT-4, GPT-3.5, and Llama-3-70B (reported in Figure 4). While GPT-4 performs best, Llama-3 achieves intermediate performance between GPT-3.5 and GPT-4 on most tasks. We believe this shows promise for deploying similar architectures with more efficient models. Additionally, the ablations results suggest that you can achieve good performance in most cases with simpler architectures with less LLM calls.
>
> **During rebuttal we have integrated more analysis of cost. We performed hyperparameter sweeps that revealed keeping just 1 hypotheses in memory (top_k=1) maintains most performance benefits while reducing costs from the default (top_k=5) Figures 11-14**. However, most of the variance in cost is drive by other factors, like episode length and performance. Notably, **higher episode costs (>$10) actually correlate with worse performance** since validating a useful hypothesis reduces subsequent LLM calls, showcasing that our method scales well when good hypotheses are generated. Our results also show robustness to other hyperparameters like learning rate and validation threshold.
>
> # Manual Preprocessing for State Representation
>
> The reviewer raises a valid point about the manual preprocessing step. While our current implementation uses manual labeling to establish a proof of concept, this step could be readily automated using either standard computer vision models or vision-language models (VLMs). For instance, a simple pretrained CNN or vision transformer classifier could be used to label the distinct visual patterns of each entity type (agents, walls, objects) in each patch. Alternatively, more recent VLMs like GPT-4V or Claude 3 Vision could also perform this classification. Both approaches would integrate with our coordinate system representation - only the patch classification step would be automated.
> Specifically, we envision a pipeline where:
>
> 1. The environment image is divided into NxM patches.
> 2. Each patch is classified using either:
>    - A trained CNN classifier optimized for this specific visual domain.
>    - A general-purpose VLM for zero-shot classification.
> 3. Classifications are mapped to our existing coordinate-based representation (e.g., `"Player Position: {'player 0-S': [(21, 4)]}"`).
>
> This approach offers several advantages:
>
> - Maintains the benefits of our structured coordinate representation which grounds spatial reasoning.
> - Can leverage either computer vision models or VLMs depending on the use case.
> - Preserves the modularity of our architecture by keeping the language interface unchanged.
> - Offers flexibility in the choice of visual processing approach based on computational constraints and accuracy requirements.
>
> # Presentation Issues
>
> We have addressed the presentation concerns:
> 1. Added clearer motivation of the core challenge in the introduction [line 30].
> 2. Updated citations to published versions where available.
> 3. Fixed the broken link in Appendix G.
> 4. Added proper figure and table links.
>
> # Hypothesis Validation Threshold
>
> We use a validation threshold of Vthr = 0.7 based on initial experiments. We have conducted additional parameter sweeps analyzing sensitivity to this threshold and included these results at [Figure 16]. The takeaways are that performance is relatively robust across different validation thresholds. It may make a bigger difference for shorter duration episodes/environments with relatively few interactions, as the key consideration is balancing between premature validation of incorrect hypotheses (if threshold is too low) and failing to validate accurate hypotheses (if too high).
>
> # Preprocessing Details
>
> We have a detailed description of the preprocessing pipeline in Appendix. Let us know if it could be more clear. Briefly we:
> 1. Divide images into NxM patches corresponding to grids of the coordinate system in the environment
> 2. Compare each patch against manually labeled reference patches for each entity type using template matching
> 3. Convert matched entities and their coordinates into a consistent text format
>
> The resulting text representation preserves all gameplay-relevant information while abstracting away irrelevant visual details. While more sophisticated visual processing could be used as mentioned above, this approach provides clean input for evaluating the core Theory of Mind mechanisms.
>
> We appreciate your thorough evaluation and your feedback has helped strengthen the paper while maintaining its key contributions.

---

> > ### Comment · Reviewer_Mz5L · 2024-11-27
> >
> > Thank you for your clarification. I will keep my positive score.

---

### Author Response · Authors · 2024-11-24
**Global Response**

# Global Response

We thank all reviewers for their thoughtful and constructive feedback. We are encouraged by all of the reviews and believe this feedback has made several substantial improvements to the manuscript:

## 1. Hyperparameter Analysis & Computational Efficiency
- Added extensive hyperparameter sweeps with 6 new figures (Figures 11-16) examining:
  - Impact of top_k parameter on performance and costs
  - Sensitivity to learning rate α and validation threshold
  - Relationship between computational cost and performance
- Key findings:
  - Performance maintained with minimal hypotheses (top_k=1) while reducing costs
  - Higher episode costs (>$10) correlate with worse performance as hypothesis validation reduces LLM calls
  - Smaller learning rates perform marginally better but the method shows robustness across different hyperparameter settings when changing learning rate α and the validation threshold
  - Included this hyperparameter and cost analysis in Appendix B

## 2. Theoretical Foundation & Related Work
- Expanded Related Work section to better situate our approach within existing literature on:
  - Chain-of-thought and Tree-of-thoughts reasoning
  - Rational Speech Act framework and Bayesian inverse planning

## 3. Improved Clarity & Motivation
- Enhanced introduction with clearer motivation of core challenges [line 30 and line 44]
- Clarified ToM module operation for multi-agent scenarios [line 230]
- Updated citations to published versions
- Fixed broken links and figure references
- Restructured ablation studies section for better readability and include hyperparameter analyses here
- Expanded conclusion with detailed discussion of limitations and future work

## 4. Model Details & Architecture
- Clarified latent variable representation in ToM module [line 200]
- Added detailed explanation of hypothesis refinement process
- Provided explicit description of LLM call structure (3 base calls + k for top-k hypotheses)
- Described separate hypothesis streams maintained for each agent in multi-agent scenarios

## 5. Limitations & Future Work
- Human in the loop: Encouraged future research for developing methods for agents to learn their prompts and appropriate types of scaffolding autonomously from environmental feedback
- Acknowledged need for broader evaluation beyond Melting Pot
- Identified opportunities for application in domains like adaptive tutoring

These revisions maintain our paper's key contributions while providing deeper analysis of our method's robustness, clearer theoretical grounding, and improved accessibility. We believe these changes significantly strengthen the manuscript and address the reviewers' main concerns.

The expanded experimental results, particularly around hyperparameter sensitivity and computational efficiency, provide important new insights about our method's practical applicability. Meanwhile, the theoretical connections to existing work in LLM reasoning and cognitive science help better contextualize our approach's novelty and potential impact.

---

### Public Comment · ~Wenqi_Zhang2 · 2024-12-02
**Interesting experiments and some recommendations**

The methods and experiments in the paper are fascinating.
I really like the idea of introducing ToM into the agent framework.

It feels like something is in common with a paper in ACL 2024.
Zhang, Wenqi, et al. "Agent-pro: Learning to evolve via policy-level reflection and optimization." arXiv preprint arXiv:2402.17574 (2024).

It is suggested that the author cite and discuss this.

---

> ### Author Response · Authors · 2024-12-03
> **Response to Wenqi Zhang**
>
> Thank you for the kind words and for bringing this paper to our attention. We will certainly discuss and cite AgentPro in the next version of our manuscript. Both papers explore policy evolution through reflection about environment, which we call hypothesis refinement here. AgentPro also generates conjectures about other agents hidden states in games with imperfect information. However, our approaches differ in important ways. Our method evaluates multiple hypotheses in parallel by making counterfactual predictions - predicting what actions agents would take under each hypothesis, even for hypotheses not actually used in gameplay. This allows us to efficiently learn from each interaction by scoring multiple possible explanations of agent behavior, and upweighting the hypothesis that fits the data the best. We will highlight these connections and differences in our revision.

---

### Meta-Review · Area_Chair_5jdo · 2024-12-20

**Metareview:**

This paper proposes a cognitively inspired method for solving multi-agent problems -- Hypothetical Minds. Most importantly, it introduces a Theory of Mind module that leverages LLMs to generate hypotheses about other agents' strategies. This method combines cognitive architectures and LLMs and achieves good performance against existing LLM-based agents and RL agents. There are concerns about the lack of generalization and limited testing scenarios. However, the contribution of this work is still significant enough to warrant an acceptance.

**Additional Comments On Reviewer Discussion:**

There is only one review leaning toward reject after the rebuttal. The concerns about generalization and limited testing are valid, but the merit of the paper outweighs the limitations.

---

### Decision · Program_Chairs · 2025-01-22

Accept (Poster)